# Bend the Basics: Degradation-Aware Deformable Tokenization for All-in-One Image Restoration

Zihao He [1]   Yunfeng Wu [1 2 3]   Xinchao Wang [4]   Songhua Liu [1]

## Abstract

All-in-one image restoration seeks a single model that can recover images degraded by diverse and spatially non-uniform corruptions. However, many unified Transformers rely on fixed patch partitioning: task/degradation condition is injected only into the backbone blocks after tokenization, leaving the embedding and reconstruction stages insensitive to local degradation variations. In contrast to previous approaches, we present **Flexible Image Transformer (FIT)** that explicitly models degradation awareness across the *entire* pipeline, from patch sampling to pixel reconstruction. Specifically, FIT employs a lightweight Degradation Encoder to predict a global degradation vector $\mathbf{g}$ and a spatial degradation map $\mathbf{M}$ from local degradation severity, which jointly condition the patch embedding and unembedding through adaptive deformation. Moreover, to improve robustness across degradation types, we introduce a task-token dropout strategy that regularizes task conditioning during training. On five standard benchmarks (BSD68, Rain100L, SOTS, GoPro, and LOLv1), FIT achieves state-of-the-art performance with 30.72 dB average PSNR on the five-degradation setting and 32.83 dB on the three-degradation setting, outperforming recent unified restoration methods by +0.5~1.1 dB. Moreover, the learned offsets provide a direct handle for visualizing degradation-aware spatial adaptation.

---

[1]School of Artificial Intelligence, Shanghai Jiao Tong University, Shanghai, China [2]Alibaba Group, Beijing, China [3]School of Advanced Technology, Xi'an Jiaotong-Liverpool University, Suzhou, China [4]Department of Electrical and Computer Engineering, National University of Singapore, Singapore. Correspondence to: Songhua Liu <liusonghua@sjtu.edu.cn>.

*Proceedings of the 43rd International Conference on Machine Learning*, Seoul, South Korea. PMLR 306, 2026. Copyright 2026 by the author(s).

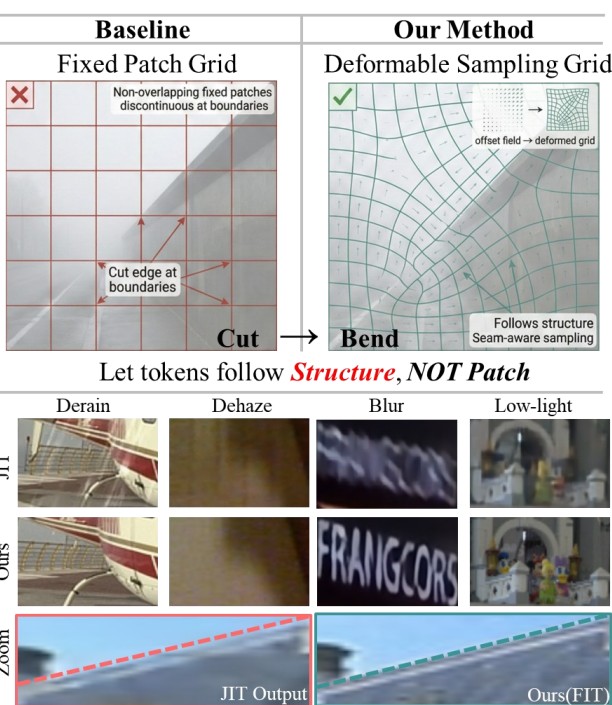

*Figure 1.* Instead of fixed patch cutting that causes boundary discontinuities, FIT bends patch grids to follow local degradation structure, enabling spatially adaptive tokenization and artifact-free restoration.

## 1. Introduction

Consider an autonomous driving scenario where a forward-facing camera captures a rainy street: the upper portion of the image is corrupted by rain streaks, while the lower portion suffers from haze-like reflections on wet pavement. A task-specific restoration model must first select a degradation label before processing, yet such spatially varying mixtures defy a single label. Applying a deraining network uniformly risks over-smoothing the haze region; applying a dehazing network may introduce color shifts in the rain region. These failures are not corner cases—real-world images routinely exhibit *spatially non-uniform* and *mixed-type* degradations that violate the single-degradation assumption underlying many restoration pipelines. All-in-one image restoration addresses this challenge by training a *single* model to handle multiple degradation types (noise, rain, haze, blur, low-light) without task-specific switching (Li

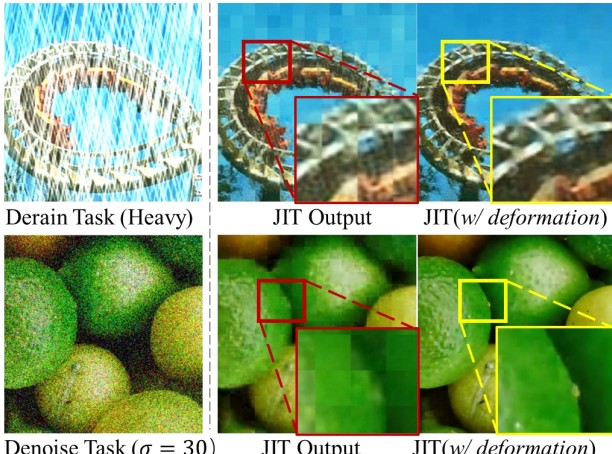

*Figure 2.* **Grid artifact analysis.** Visual comparison on derain and denoise tasks shows JIT produces visible grid artifacts at patch boundaries (red boxes), while deformable tokenization mitigates this issue (yellow boxes).

et al., 2022; Potlapalli et al., 2023; Zhang et al., 2023). The key difficulty is to design a unified architecture that can *adapt its processing locally* to heterogeneous degradations, rather than applying a spatially homogeneous operation everywhere. This requirement motivates a closer examination of how existing unified methods inject degradation information, and where they still fall short.

Existing unified restoration methods can be roughly grouped into three categories. (A) Task-specific prompts or tokens prepend learnable embeddings to signal the degradation type (Potlapalli et al., 2023; Conde et al., 2024); the Transformer then conditions its attention on these tokens. (B) Global degradation embeddings encode the input into a holistic vector that modulates intermediate features via FiLM or cross-attention (Li et al., 2022; Zhang et al., 2023; Cui et al., 2025). (C) Multi-branch architectures allocate task-specific heads or sub-networks (Valanarasu et al., 2022; Li et al., 2020), improving capacity at the cost of parameter redundancy and limited sharing.

A recent direct-prediction Transformer, **JIT** (*Just image Transformers*) (Li & He, 2025), shows that image generation can be simplified by letting the model predict clean images directly. When adapted as a restoration baseline, JIT retains fixed patch tokenization and reconstruction, leaving *tokenization and reconstruction* tied to fixed patch grids. This design choice introduces a critical limitation: fixed $P \times P$ patch boundaries often cut through degradation structures (e.g., rain streaks, blur kernels), causing the model to mix corrupted and clean pixels within the same token. During reconstruction, these mixed tokens are mapped back to a rigid grid, producing visible **grid artifacts** at patch boundaries, a phenomenon we illustrate in Fig. 2. As degradation severity increases, these artifacts become more pronounced, ultimately bottlenecking restoration quality regardless of model capacity.

Focusing on these drawbacks, we argue in this paper that degradation-aware processing should span the *entire* pipeline, from patch sampling, through token transformation, to pixel reconstruction. This suggests a simple but under-explored principle: make patch boundaries and reconstruction functions of local degradation severity. Building on this insight, we propose Flexible Image Transformer (FIT), a unified restoration Transformer with degradation-aware *deformable* tokenization (Fig. 1).

Specifically, FIT employs a lightweight Degradation Encoder to predict a global degradation vector $\mathbf{g}$ and a spatial degradation map $\mathbf{M}$ from the input image. Conditioned on $(\mathbf{g}, \mathbf{M})$, FIT performs FiLM-modulated deformable patch embedding and unembedding: the embedding stage learns content-adaptive sampling offsets to align patch boundaries with local corruption patterns, and the unembedding stage inversely warps tokens back to pixels with the same conditioning, ensuring smooth reconstruction without grid discontinuities. To reduce reliance on explicit task labels, we further introduce a task-token dropout strategy that stochastically replaces hard task tokens with soft tokens derived from $\mathbf{g}$, encouraging content-driven degradation inference.

Our contributions are summarized as follows:

- We propose **FIT**, a unified image restoration architecture that makes the pipeline end-to-end degradation-aware by conditioning deformable embedding and reconstruction on global-local degradation representations.
- We develop a task-token dropout strategy to regularize task conditioning and improve robustness to spatially mixed or unseen degradation combinations.
- We demonstrate state-of-the-art performance on five restoration tasks (denoising, deraining, dehazing, deblurring, low-light enhancement) across standard benchmarks including BSD68, Rain100L, SOTS, GoPro, and LOLv1. The learned offsets also provide an intuitive handle to visualize degradation-adaptive spatial sampling.

**Conflict of Interest Disclosure.** The authors declare no financial or other substantive conflicts of interest.

## 2. Related Work

### 2.1. Unified All-in-One Restoration Frameworks

All-in-one image restoration aims to handle multiple degradations using a single model, avoiding task-specific switching and enabling parameter sharing. AirNet (Li et al., 2022) introduces contrastive degradation learning to build a unified backbone without explicit task labels. IDR (Zhang et al., 2023) reformulates degradations as combinations of basic "ingredients" for improved generalization. PromptIR (Potlapalli et al., 2023) adopts learnable prompts to

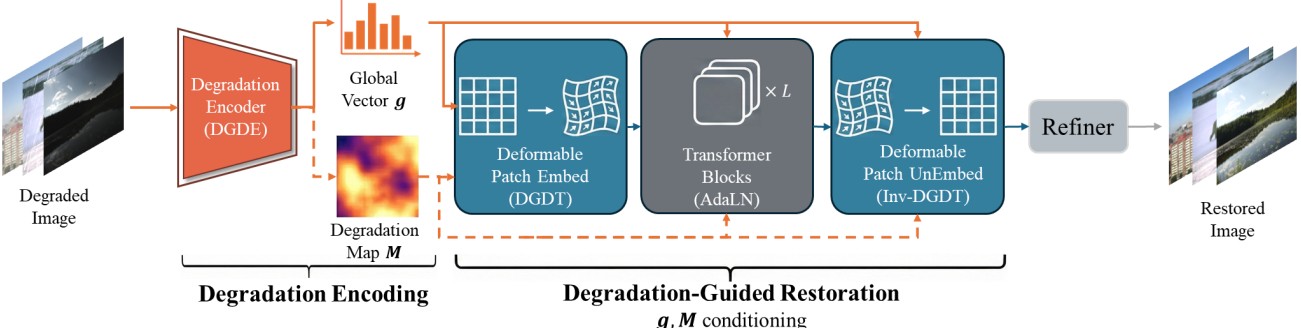

*Figure 3.* **Overview of FIT.** Given a degraded image, the Degradation Encoder extracts a global vector **g** and spatial map M. Conditioned on (**g**, M), deformable patch embedding samples pixels with learned offsets, the AdaLN Transformer processes tokens with **g**-modulation, and deformable unembedding reconstructs pixels via inverse warping. A lightweight refiner produces the final output.

guide task-specific processing. More recent works further advance this direction: InstructIR (Conde et al., 2024) enables natural language instructions, DA-CLIP (Luo et al., 2024) leverages vision-language alignment for zero-shot generalization, AdaIR (Cui et al., 2025) proposes adaptive frequency mining, and VLU-Net (Zeng et al., 2025) combines visual-language understanding. The recent JIT formulation (Li & He, 2025) further provides a strong direct-prediction Transformer baseline, but its image-to-token and token-to-image mappings still rely on fixed patch grids. Despite these advances, most frameworks perform tokenization with fixed patch grids, while degradation cues only modulate representations during token processing (Zamir et al., 2022). This can be suboptimal for spatially non-uniform degradations. This limitation is related to deformable operations: deformable convolutions and efficient dynamic sparse operators learn offsets for adaptive aggregation (Dai et al., 2017; Zhu et al., 2019; Xiong et al., 2024), while deformable attention and recent Transformer/SSM variants introduce data-dependent sparse sampling or routing inside backbone aggregation (Xia et al., 2022; BaoLong et al., 2024; Liu et al., 2025). Unlike these general feature extraction modules, FIT conditions offsets on degradation representations and applies deformation at the image-token interface itself, during patch embedding and unembedding, to address restoration-specific grid discontinuities. FIT extends degradation awareness to tokenization and reconstruction, enabling end-to-end spatially adaptive processing.

### 2.2. Task-Specific Prompts and Tokens

A common strategy for unified restoration is to introduce task identity through learnable *task tokens* or *prompt embeddings* (Jia et al., 2022). PromptIR (Potlapalli et al., 2023) learns task-specific prompts that modulate attention computation. ProRes (Ma et al., 2023) explores prompt-based restoration with progressive refinement. InstructIR (Conde et al., 2024) enables text-based instructions for flexible task specification. Beyond explicit prompts, some methods learn implicit task embeddings: TAPE (Liu et al., 2022) uses

task-agnostic prior estimation, and Diff-Plugin (Liu et al., 2024) introduces pluggable task-specific modules. However, explicit task identifiers provide only coarse-grained guidance and may be less informative under mixed or spatially varying degradations(Zhang et al., 2026; Luo et al., 2025). Moreover, token-based conditioning acts during token transformation rather than changing how pixels are partitioned. FIT uses task tokens but additionally makes tokenization and reconstruction degradation-aware, enabling spatially adaptive processing beyond token transformation.

### 2.3. Degradation Representation Learning

Another line of work infers degradation characteristics directly from the input, reducing the need for task labels at inference. These methods learn degradation representations that modulate features via FiLM (Perez et al., 2018), AdaIN (Huang & Belongie, 2017), or cross-attention. Air-Net (Li et al., 2022) learns contrastive embeddings that cluster similar degradations. IDR (Zhang et al., 2023) decomposes degradations into learnable ingredients. DA-CLIP (Luo et al., 2024) aligns degradation representations with CLIP's semantic space. AdaIR (Cui et al., 2025) learns frequency-aware embeddings for spectral modulation. A key challenge is *granularity*: global vectors provide image-level conditioning but cannot capture *where* corruption is severe within a single image. Some works predict spatially varying maps (Zamir et al., 2022; Chen et al., 2023), but these typically modulate intermediate features rather than tokenization. FIT uses both global vectors and spatial maps to guide deformable tokenization, enabling location-dependent adaptation at the embedding boundary.

## 3. Method

### 3.1. Overview

Given a degraded image $\mathbf{x} \in \mathbb{R}^{H \times W \times 3}$, FIT recovers a clean image $\hat{\mathbf{y}}$ through four stages (Fig. 3): (1) a Dual-Granularity Degradation Encoder (DGDE) extracts global

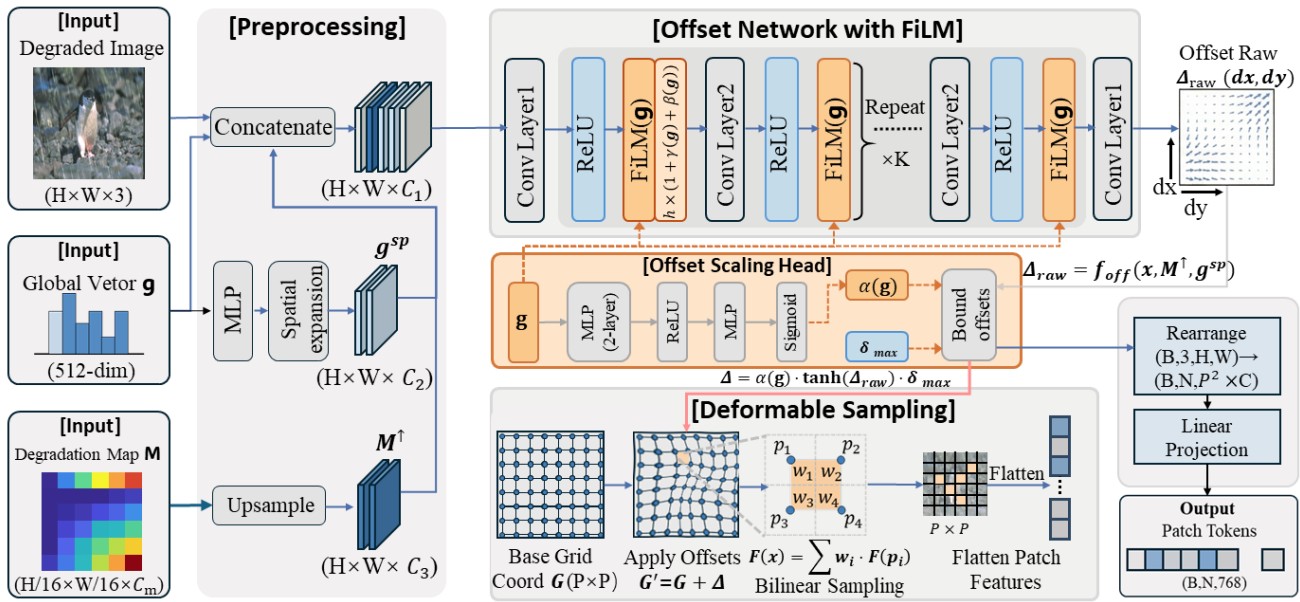

*Figure 4.* Conditional Deformable Patch Embedding. The offset network takes concatenated inputs $(\mathbf{x}, \mathbf{M}^{\uparrow}, \mathbf{g}^{\text{sp}})$ and predicts sampling offsets via FiLM-modulated convolutions. Dynamic scaling adjusts offset magnitude based on degradation severity.

vector $\mathbf{g}$ and spatial map $\mathbf{M}$; (2) Degradation-Guided Deformable Tokenization (DGDT) converts pixels to tokens with learned sampling offsets; (3) an AdaLN Transformer processes tokens conditioned on $\mathbf{g}$; (4) inverse DGDT reconstructs pixels, followed by a lightweight refiner. A Task-Token Dropout (TTD) strategy improves robustness when task labels are unavailable.

### 3.2. Dual-Granularity Degradation Encoder

Effective unified restoration requires understanding *what* degradation is present and *where* it is severe. Prior degradation encoders typically produce only a global vector, which cannot express spatial variation within a single image, a critical limitation when degradations are spatially non-uniform. DGDE addresses this by producing two complementary representations: a global vector $\mathbf{g} \in \mathbb{R}^{D_g}$ encoding image-wide statistics, and a spatial degradation map $\mathbf{M} \in \mathbb{R}^{C_m \times h \times w}$ (where $h = H/16$, $w = W/16$) capturing location-dependent severity.

The degraded image $\mathbf{x}$ is processed by a lightweight CNN backbone $f_{\text{enc}}$ with strided convolutions ($16\times$ downsampling) to obtain feature $\mathbf{F}$. The degradation map is computed as $\mathbf{M} = \tanh(\text{Conv}_{1\times1}(\mathbf{F}))$, where $\tanh$ normalizes values to $[-1, 1]$ for stable conditioning. The global vector is obtained by $\mathbf{g} = \text{MLP}(\text{GAP}(\mathbf{F}))$, where GAP denotes global average pooling.

To encourage semantically meaningful representations, two auxiliary heads supervise $\mathbf{g}$ during training:

$$\mathcal{L}_{\text{aux}} = \text{CE}(\text{Linear}(\mathbf{g}), t) + \lambda_s \|\sigma(\text{MLP}(\mathbf{g})) - s\|^2, \quad (1)$$

where $t$ and $s$ denote ground-truth degradation type and strength, respectively.

### 3.3. Degradation-Guided Deformable Tokenization

Standard ViTs partition images into fixed $P \times P$ patches regardless of content, which can mix corrupted and clean regions within a single token. DGDT learns content-adaptive sampling offsets conditioned on $(\mathbf{g}, \mathbf{M})$, allowing patch boundaries to align with local corruption structure (Fig. 4).

**Deformable Patch Embedding.** We first upsample $\mathbf{M}$ to input resolution as $\mathbf{M}^{\uparrow}$ and expand $\mathbf{g}$ spatially as $\mathbf{g}^{\text{sp}}$. These are concatenated with $\mathbf{x}$ and fed to an offset network $f_{\text{off}}$, which consists of strided convolutions interleaved with FiLM layers (Perez et al., 2018) that modulate features using $\mathbf{g}$:

$$\text{FiLM}(\mathbf{h}, \mathbf{g}) = \mathbf{h} \odot (1 + \gamma(\mathbf{g})) + \beta(\mathbf{g}), \quad (2)$$

where $\gamma, \beta$ are learned linear projections initialized to zero for stable training. The network outputs offset map $\boldsymbol{\Delta} \in \mathbb{R}^{2 \times H \times W}$, normalized and dynamically scaled:

$$\boldsymbol{\Delta} = \alpha(\mathbf{g}) \cdot \delta_{\max} \cdot \tanh(f_{\text{off}}(\mathbf{x}, \mathbf{M}^{\uparrow}, \mathbf{g}^{\text{sp}})), \quad (3)$$

where $\alpha(\mathbf{g}) = 0.5 + \sigma(\text{MLP}(\mathbf{g}))$ adaptively adjusts offset magnitude based on degradation severity, and $\delta_{\max}$ is a hyperparameter bounding maximum displacement. Pixels are sampled at offset locations via bilinear interpolation, then reshaped into patches and projected to tokens $\mathbf{Z}^{(0)} \in \mathbb{R}^{N \times D}$.

**Deformable Patch Unembedding.** After Transformer processing, tokens $\mathbf{Z}^{(L)}$ are projected back to pixel space

as $\tilde{\mathbf{y}}$. Inverse offsets $\boldsymbol{\Delta}'$ are predicted using a similar FiLM-modulated network, and the reconstruction is warped back to the original grid. A lightweight CNN refiner with residual connection produces the final output $\hat{\mathbf{y}} = \tilde{\mathbf{y}}_{\text{warp}} + f_{\text{refine}}(\tilde{\mathbf{y}}_{\text{warp}})$.

### 3.4. Task-Token Dropout and Transformer Backbone

While DGDE provides implicit degradation conditioning, explicit task guidance can further improve performance when degradation types are known. FIT maintains a bank of *hard* task tokens $\mathbf{T}_{\text{hard}} \in \mathbb{R}^{K \times N_t \times D}$ indexed by task type, alongside a projection that generates *soft* tokens from $\mathbf{g}$:

$$\mathbf{T}_{\text{soft}} = \text{Reshape}(\text{Linear}(\mathbf{g})) \in \mathbb{R}^{N_t \times D}. \quad (4)$$

This dual-mode design allows FIT to operate with or without explicit task labels.

To prevent over-reliance on oracle labels and bridge the train-test gap, Task-Token Dropout (TTD) stochastically switches between modes during training:

$$\mathbf{T} = (1-m) \cdot \mathbf{T}_{\text{hard}}[t] + m \cdot \mathbf{T}_{\text{soft}}, \quad m \sim \text{Bernoulli}(p_{\text{drop}}), \quad (5)$$

where $p_{\text{drop}} = 0.3$ empirically balances label utilization and label-free inference capability (see Sec. 4.3). When dropout is triggered, the model must infer task-relevant information from $\mathbf{g}$, encouraging robust degradation representations.

Beyond task tokens, spatial conditioning is injected by projecting $\mathbf{M}$ and adding it to patch tokens after positional embedding. The augmented sequence is processed by $L$ AdaLN Transformer blocks(Huang & Belongie, 2017), where each layer computes six modulation parameters from $\mathbf{g}$:

$$(\alpha_1, \beta_1, \gamma_1, \alpha_2, \beta_2, \gamma_2)^{(l)} = \text{MLP}^{(l)}(\mathbf{g}), \quad (6)$$

which scale, shift, and gate the attention and feed-forward outputs. All modulation parameters are initialized to zero, ensuring stable training from a standard Transformer baseline.

### 3.5. Training Objectives

FIT is trained with a weighted combination of losses:

$$\mathcal{L} = \mathcal{L}_{\text{recon}} + \lambda_t \mathcal{L}_{\text{type}} + \lambda_s \mathcal{L}_{\text{strength}} + \lambda_b \mathcal{L}_{\text{seam}} + \lambda_o \mathcal{L}_{\text{off}}. \quad (7)$$

The reconstruction loss $\mathcal{L}_{\text{recon}} = \|\hat{\mathbf{y}} - \mathbf{y}\|_1$ measures pixel-wise fidelity. The type classification loss $\mathcal{L}_{\text{type}}$ (cross-entropy) and strength regression loss $\mathcal{L}_{\text{strength}}$ (MSE, applied only to samples with known strength labels) supervise the auxiliary heads. Here $t$ is an auxiliary type label, and $s = (q - q_{\text{min}})/(q_{\text{max}} - q_{\text{min}}) \in [0, 1]$ normalizes each task's generation parameter $q$ (noise, rain, haze, blur, or illumination). Both are training-only; at test time FIT needs no $t/s$, and mixed cases rely on image-derived $(\mathbf{g}, \mathbf{M})$.

---

**Algorithm 1** FIT Forward Pass

**Require:** Degraded image $\mathbf{x} \in \mathbb{R}^{H \times W \times 3}$; optional task label $t$
**Ensure:** Restored image $\hat{\mathbf{y}} \in \mathbb{R}^{H \times W \times 3}$
1: $\mathbf{g}, \mathbf{M} \leftarrow \text{DGDE}(\mathbf{x})$
2: $\boldsymbol{\Delta} \leftarrow f_{\text{off}}(\mathbf{x}, \mathbf{M}, \mathbf{g})$
3: $\tilde{\mathbf{x}} \leftarrow \text{GridSample}(\mathbf{x}, \boldsymbol{\Delta})$
4: $\mathbf{Z} \leftarrow \text{PatchEmbed}(\tilde{\mathbf{x}})$
5: $\mathbf{T} \leftarrow \text{TTD}(\mathbf{g}, t, p_{\text{drop}})$
6: $\mathbf{Z} \leftarrow [\mathbf{T}; \mathbf{Z}] + \mathbf{E}_{\text{pos}}$
7: $\mathbf{Z}_{N_t:} \leftarrow \mathbf{Z}_{N_t:} + \text{Linear}(\mathbf{M})$
8: **for** $l = 1, \dots, L$ **do**
9: $\quad \mathbf{Z} \leftarrow \text{AdaLNBlock}^{(l)}(\mathbf{Z}, \mathbf{g})$
10: **end for**
11: $\tilde{\mathbf{y}} \leftarrow \text{PatchUnEmbed}(\mathbf{Z}_{N_t:})$
12: $\boldsymbol{\Delta}' \leftarrow f'_{\text{off}}(\tilde{\mathbf{y}}, \mathbf{M}, \mathbf{g})$
13: $\hat{\mathbf{y}} \leftarrow \text{Refine}(\text{GridSample}(\tilde{\mathbf{y}}, -\boldsymbol{\Delta}'))$

---

For seam-aware training, let $\mathcal{B}_v$ and $\mathcal{B}_h$ denote adjacent pixel pairs crossing vertical and horizontal patch boundaries, respectively. For each pair $(p, q)$, define $d_{\hat{\mathbf{y}}}(p, q) = \hat{\mathbf{y}}_p - \hat{\mathbf{y}}_q$ and $d_{\mathbf{y}}(p, q) = \mathbf{y}_p - \mathbf{y}_q$. With $\eta = 0.5$, the seam loss is

$$\mathcal{L}_{\text{seam}} = \frac{1}{2} \sum_{r \in \{v,h\}} \frac{1}{|\mathcal{B}_r|} \sum_{(p,q) \in \mathcal{B}_r} \ell(p, q),$$
$$\ell(p, q) = \|d_{\hat{\mathbf{y}}}(p, q) - d_{\mathbf{y}}(p, q)\|_1 + \eta \|d_{\hat{\mathbf{y}}}(p, q)\|_1, \quad (8)$$

The first term matches ground-truth cross-boundary gradients, while the second discourages seam-like jumps introduced by patch partitioning. The offset regularization $\mathcal{L}_{\text{off}} = \|\boldsymbol{\Delta}\|_2^2 + \mu \|\nabla \boldsymbol{\Delta}\|_2^2$ encourages small, spatially smooth offsets to prevent degenerate sampling.

## 4. Experiments

### 4.1. Experimental Setup

**Datasets.** We evaluate FIT on five image restoration tasks using widely adopted benchmarks. For **denoising**, we train on DIV2K (Agustsson & Timofte, 2017) with synthetic Gaussian noise ($\sigma \in \{15, 25, 50\}$) and test on BSD68 (Martin et al., 2001). For **deraining**, we use Rain100L (Yang et al., 2017) containing paired rain/clean images with varying rain streak densities. For **dehazing**, we train on RESIDE-ITS (Li et al., 2018) and evaluate on SOTS dataset. For **deblurring**, we adopt the GoPro dataset (Nah et al., 2017) with 2,103 training and 1,111 test image pairs exhibiting motion blur. For **low-light enhancement**, we use LOLv1 (Wei et al., 2018) and LOLv2 (Yang et al., 2021) containing paired low/normal-light images. Following prior unified restoration works (Li et al., 2022; Potlapalli et al., 2023), all models are trained jointly on the combined training sets.

**Implementation Details.** FIT uses hidden dimension $D = 896$, $L = 24$ Transformer blocks, $N_h = 14$ attention

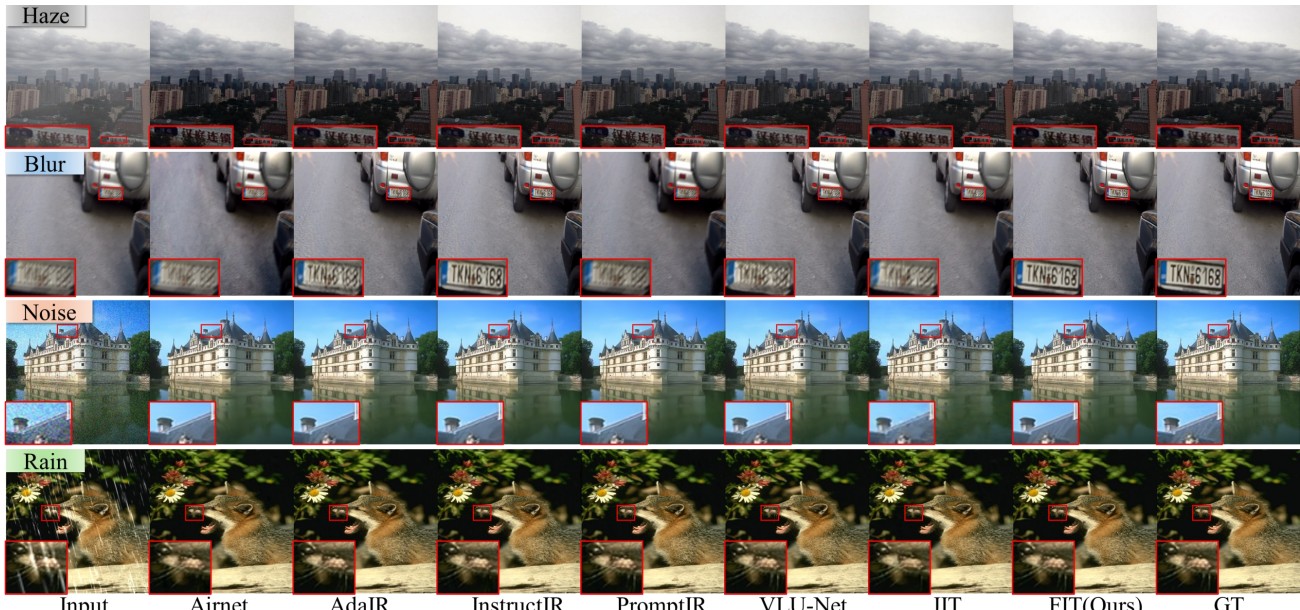

*Figure 5.* Qualitative comparison on four degradation types: deraining, dehazing, denoising, and deblurring. Red boxes highlight zoomed-in regions for detailed comparison. Our FIT produces cleaner textures, sharper edges, and more faithful color restoration compared to existing methods.

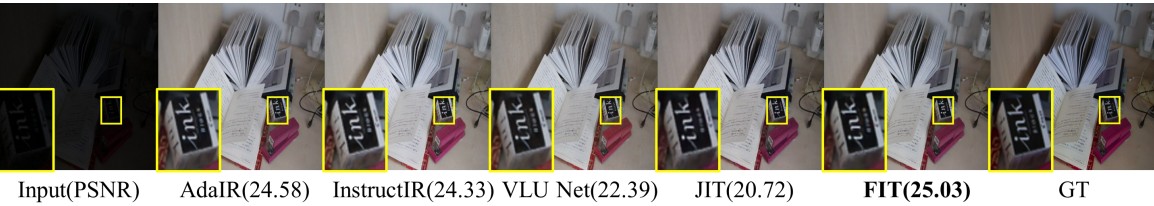

*Figure 6.* Qualitative comparison on LOLV1 dataset. Yellow boxes highlight zoomed-in regions for detailed comparison.

heads, degradation embedding dimension $D_g = 768$, and spatial map channels $C_m = 128$. All models use patch size $P = 16$ and $N_t = 4$ task tokens with dropout probability $p_{\mathrm{drop}} = 0.3$. Training is performed on 1 NVIDIA A100 GPU with batch size 64 and gradient accumulation of 4 (effective batch size 256). We use AdamW optimizer (Loshchilov & Hutter, 2019) with $\beta_1 = 0.9$, $\beta_2 = 0.95$, learning rate $6 \times 10^{-5}$ with cosine decay to $10^{-6}$, and weight decay 0.05. We train for 1500 epochs with mixed-precision (BF16). Loss weights are set to $\lambda_t = 0.1$, $\lambda_s = 0.05$, $\lambda_b = 0.3$, $\lambda_o = 0.01$. Input images are randomly cropped to $256 \times 256$ during training.

**Baselines.** We compare FIT against recent unified restoration methods: DL (Fan et al., 2019) (decoupled learning), AirNet (Li et al., 2022) (contrastive degradation embedding), IDR (Zhang et al., 2023) (ingredient reformulation), PromptIR (Potlapalli et al., 2023) (learnable prompts), NDR (Yao et al., 2024) (neural degradation representation), Gridformer (Wang et al., 2024) (grid attention), InstructIR (Conde et al., 2024) (language instructions), Perceive-IR (Zhang et al., 2025) (perceptual guidance), AdaIR (Cui et al., 2025) (frequency modulation), and VLU-Net (Zeng

et al., 2025) (visual-language understanding). We use JIT (Li & He, 2025) as primary baseline since FIT extends it with deformable tokenization. Results are reported on *three-degradation* (Table 1) and *five-degradation* (Table 3) settings.

### 4.2. Comparison with State-of-the-Art

**Quantitative Results.** Tables 1 and 3 present comprehensive comparisons on the three-degradation and five-degradation settings, respectively. FIT achieves state-of-the-art performance on both settings.

On the three-degradation setting (Table 1), FIT obtains 32.83 dB average PSNR, outperforming the previous best method VLU-Net by +0.13 dB and the JIT baseline by +1.11 dB. Notably, FIT achieves the best results on dehazing (31.11 dB) and denoising across all noise levels, demonstrating the effectiveness of degradation-aware deformable tokenization. On the more challenging five-degradation setting (Table 3), FIT achieves 30.72 dB average PSNR, surpassing AdaIR by +0.52 dB and the JIT baseline by a significant margin of +1.44 dB. The improvement is particularly pronounced on

*Table 1.* Comparison to state-of-the-art AiOIR methods on Three Degradations task. **Bold** indicates the best result and underline indicates the second best.

| Method | Source | Dehazing | | Deraining | | Denoising | | | | | | Average | |
| | | SOTS | | Rain100L | | BSD68$_{\sigma=15}$ | | BSD68$_{\sigma=25}$ | | BSD68$_{\sigma=50}$ | | | |
|---|---|---|---|---|---|---|---|---|---|---|---|---|---|
| DL | TPAMI'19 | 26.92 | 0.931 | 32.62 | 0.931 | 33.05 | 0.914 | 30.41 | 0.861 | 26.90 | 0.740 | 29.98 | 0.876 |
| AirNet | CVPR'22 | 27.94 | 0.962 | 34.90 | 0.967 | 33.92 | 0.933 | 31.26 | 0.888 | 28.00 | 0.797 | 31.20 | 0.910 |
| IDR | CVPR'23 | 29.87 | 0.970 | 36.03 | 0.971 | 33.89 | 0.931 | 31.32 | 0.884 | 28.04 | 0.798 | 31.83 | 0.911 |
| PromptIR | NeurIPS'23 | 30.58 | 0.974 | 36.37 | 0.972 | 33.98 | 0.933 | 31.31 | 0.888 | 28.06 | 0.799 | 32.06 | 0.913 |
| NDR | TIP'24 | 28.64 | 0.962 | 35.42 | 0.969 | 34.01 | 0.932 | 31.36 | 0.887 | 28.10 | 0.798 | 31.51 | 0.910 |
| Gridformer | IJCV'24 | 30.37 | 0.970 | 37.15 | 0.972 | 33.93 | 0.931 | 31.37 | 0.887 | 28.11 | 0.801 | 32.19 | 0.912 |
| InstructIR | ECCV'24 | 30.22 | 0.959 | 37.98 | 0.978 | 34.15 | 0.933 | 31.52 | 0.890 | 28.30 | 0.803 | 32.43 | 0.913 |
| Perceive-IR | TIP'25 | 30.87 | 0.975 | 38.29 | 0.980 | 34.13 | 0.934 | 31.53 | 0.890 | 28.31 | 0.804 | 32.63 | 0.917 |
| AdaIR | ICLR'25 | 31.06 | 0.980 | 38.64 | 0.983 | 34.12 | 0.934 | 31.45 | 0.892 | 28.19 | 0.802 | 32.69 | 0.918 |
| VLU-Net | CVPR'25 | 30.71 | 0.980 | **38.93** | **0.984** | 34.13 | 0.935 | 31.48 | 0.892 | 28.23 | 0.804 | 32.70 | 0.919 |
| JIT(baseline) | - | 30.11 | 0.954 | 35.54 | 0.968 | 33.77 | 0.928 | 31.17 | 0.878 | 28.03 | 0.797 | 31.72 | 0.905 |
| **FIT (Ours)** | - | **31.11** | **0.981** | 38.92 | **0.984** | **34.23** | **0.936** | **31.55** | **0.894** | **28.32** | **0.806** | **32.83** | **0.920** |

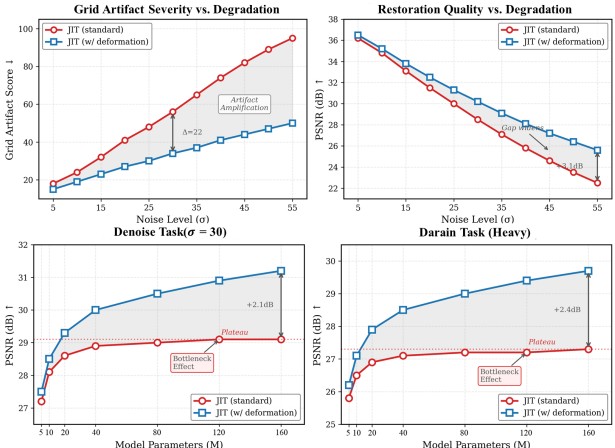

*Figure 7.* Grid artifact bottleneck analysis. Top: artifact severity and PSNR vs. degradation level. Bottom: model scaling performance on denoise ($\sigma$=30) and derain (heavy) tasks. Standard JIT plateaus due to grid artifacts, while deformable tokenization enables continued scaling with up to +2.4dB improvement.

tasks with spatially varying degradations: deraining (+1.66 dB vs JIT), dehazing (+2.52 dB vs JIT), and deblurring (+1.10 dB vs JIT). This validates that extending degradation conditioning to the tokenization boundary is especially beneficial when degradations are spatially non-uniform.

**Qualitative Results.** Figure 5 shows visual comparisons on challenging examples across four degradation types. On hazy images (first row), methods with global-only conditioning such as AirNet struggle with depth-varying haze, whereas FIT's spatial degradation map enables location-aware dehazing with better color fidelity. On motion-blurred images (second row), the deformable tokenization helps FIT recover sharper edges and finer textures compared to meth-

*Table 2.* Core ablation study of the proposed components. Deform denotes deformable tokenization, Deg-Cond denotes degradation-aware conditioning, Seam denotes seam-aware training, and TTD denotes task-token dropout.

| ID | Deform | Deg-Cond | Seam | TTD | PSNR↑ | Grid Score↓ |
|---|---|---|---|---|---|---|
| A | ✗ | ✗ | ✗ | ✗ | 26.81 | 52.4 |
| B | ✓ | ✗ | ✗ | ✗ | 27.35 | 46.9 |
| C | ✓ | ✓ | ✗ | ✗ | 27.92 | 44.1 |
| D | ✓ | ✓ | ✓ | ✗ | 28.25 | 38.7 |
| E | ✓ | ✓ | ✓ | ✓ | **28.32** | **36.3** |

ods using fixed patch grids. On noisy images (third row), FIT produces cleaner results while preserving structural details that other methods tend to over-smooth. On rainy images (fourth row), FIT adaptively processes different regions, removing rain streaks more thoroughly while preserving background textures. Appendix Figure 10 further provides enlarged boundary visualizations and patch-boundary gradient maps: JIT exhibits bright seam responses along fixed patch borders, while FIT suppresses these discontinuities and yields lower Grid Scores.

Figure 6 presents additional comparisons on the low-light enhancement task. FIT achieves the highest PSNR (25.03 dB) while producing visually pleasing results with accurate color restoration and minimal noise amplification.

### 4.3. Ablation Studies

We conduct ablation experiments to analyze the contribution of each component. All variants are trained for same epochs under identical settings.

**Component-wise Analysis.** Table 2 presents an incremental denoising ablation. Starting from a vanilla Transformer

*Table 3.* Comparison to state-of-the-art AiOIR methods on Five Degradations task. **Bold** indicates the best result and underline indicates the second best.

| Method | Source | Dehazing | | Deraining | | Denoising | | Deblurring | | Low-Light | | Average | |
|---|---|---|---|---|---|---|---|---|---|---|---|---|---|
| | | SOTS | | Rain100L | | BSD68$_{\sigma=25}$ | | GoPro | | LOLv1 | | | |
| DL | TPAMI'19 | 20.54 | 0.826 | 21.96 | 0.762 | 23.09 | 0.745 | 19.86 | 0.672 | 19.83 | 0.712 | 21.05 | 0.743 |
| AirNet | CVPR'22 | 21.04 | 0.884 | 32.98 | 0.951 | 30.91 | 0.882 | 24.35 | 0.781 | 18.18 | 0.735 | 25.49 | 0.847 |
| IDR | CVPR'23 | 25.24 | 0.943 | 35.63 | 0.965 | **31.60** | 0.887 | 27.87 | 0.846 | 21.34 | 0.826 | 28.34 | 0.893 |
| PromptIR | NeurIPS'23 | 26.54 | 0.949 | 36.37 | 0.970 | 31.47 | 0.886 | 28.71 | 0.881 | 22.68 | 0.832 | 29.15 | 0.904 |
| Gridformer | IJCV'24 | 26.79 | 0.951 | 36.61 | 0.971 | 31.45 | 0.885 | 29.22 | 0.884 | 22.59 | 0.831 | 29.33 | 0.904 |
| InstructIR | ECCV'24 | 27.10 | 0.956 | 36.84 | 0.973 | 31.40 | 0.887 | 29.40 | 0.886 | 23.00 | 0.836 | 29.55 | 0.907 |
| Perceive-IR | TIP'25 | 28.19 | 0.964 | 37.25 | 0.977 | 31.44 | 0.887 | 29.46 | 0.886 | 22.81 | 0.833 | 29.84 | 0.909 |
| AdaIR | ICLR'25 | 30.53 | 0.978 | 38.02 | 0.981 | 31.35 | 0.888 | 28.12 | 0.858 | 23.00 | 0.845 | 30.20 | 0.910 |
| VLU-Net | CVPR'25 | **30.84** | **0.980** | 38.54 | 0.982 | 31.43 | 0.891 | 27.46 | 0.840 | 22.29 | 0.833 | 30.11 | 0.905 |
| JIT(baseline) | - | 28.29 | 0.964 | 36.90 | 0.975 | 30.04 | 0.879 | 28.42 | 0.879 | 22.77 | 0.832 | 29.28 | 0.906 |
| **Ours** | - | 30.81 | 0.979 | **38.56** | **0.983** | 31.57 | **0.893** | **29.52** | **0.890** | **23.13** | **0.848** | **30.72** | **0.919** |

*Table 4.* Cumulative component ablations beyond denoising. Values are PSNR.

| Task | +Deform | +Deg-Cond | +Seam | Full FIT |
|---|---|---|---|---|
| Derain (Rain100L) | 37.41 | 38.02 | 38.34 | **38.56** |
| Dehaze (SOTS) | 29.98 | 30.34 | 30.55 | **30.81** |
| Deblur (GoPro) | 28.87 | 29.12 | 29.31 | **29.52** |

*Table 5.* Ablation on Task-Token Dropout (TTD). Values are PSNR.

| Setting | Hard | Soft | Wrong | Mixed |
|---|---|---|---|---|
| w/o TTD | 32.88 | 32.31 | 32.07 | 30.15 |
| TTD $p=0.1$ | 32.86 | 32.34 | 32.15 | 30.48 |
| TTD $p=0.3$ | 32.83 | 32.33 | **32.25** | **30.84** |

*Table 6.* Control study on synthetic mixed degradations. Values are PSNR.

| Variant | Avg. | Dominant | Boundary | Light |
|---|---|---|---|---|
| Content-adaptive only | 27.83 | 27.21 | 25.84 | 30.62 |
| w/o spatial map $\mathbf{M}$ | 28.12 | 27.46 | 26.09 | 30.73 |
| Full FIT (g+M) | **28.96** | **28.31** | **27.12** | **30.96** |

suggests that randomly dropping task tokens during training encourages the restoration network to rely more on the inferred degradation representation $\mathbf{g}$ and spatial map $\mathbf{M}$, improving robustness under ambiguous inputs.

**Sensitivity to Offset Bound and Patch Size.** We further test three offset bounds ($0.125P$, $0.25P$, and $0.375P$) and three patch sizes (8, 16, and 32), obtaining 5-task averages of (30.41, 30.72, 30.70) and (30.66, 30.72, 30.21), respectively. Too small an offset bound limits boundary adaptation, while overly large patches reduce local flexibility and aggravate grid artifacts; therefore we use $\delta_{\max} = 0.25P$ and $P = 16$.

**Mixed-Degradation Control Study.** To test whether offsets encode degradation-aware structure rather than image content, we synthesize spatially heterogeneous inputs from three composable degradations (Gaussian noise, motion blur, low-light): per image, two are sampled—one applied globally, the other through a random smooth mask, yielding known regions with exact ground truth. We compare a content-adaptive baseline, FIT with $\mathbf{g}$ only, and full FIT ($\mathbf{g+M}$), reporting PSNR on the dominant region, boundary band, and light region (no oracle label at inference). Table 6 shows full FIT's gains concentrate in dominant and boundary regions while light-region gains are small, confirming $\mathbf{M}$ encodes local degradation structure.

baseline (Row A, 26.81 dB), adding deformable tokenization improves PSNR by +0.54 dB while reducing the Grid Score from 52.4 to 46.9. Incorporating degradation-aware conditioning adds a further +0.57 dB, showing that degradation cues guide offsets better than content-only adaptation. Seam-aware training improves both PSNR and Grid Score by explicitly suppressing patch-boundary discontinuities, and TTD yields the full model at 28.32 dB with the lowest Grid Score of 36.3. Table 4 confirms the same trend on deraining, dehazing, and deblurring, indicating that the gains are not limited to denoising.

**Effect of Task-Token Dropout.** Table 5 examines hard-, soft-, wrong-token, and mixed-degradation inference. Without TTD, the model performs well when ground-truth task labels are provided, but degrades when using soft or incorrect tokens, which simulates unknown degradation metadata at test time. TTD keeps hard-token performance stable, reduces the wrong-token drop from 0.81 to 0.58 dB, and improves mixed degradation from 30.15 to 30.84 dB. This

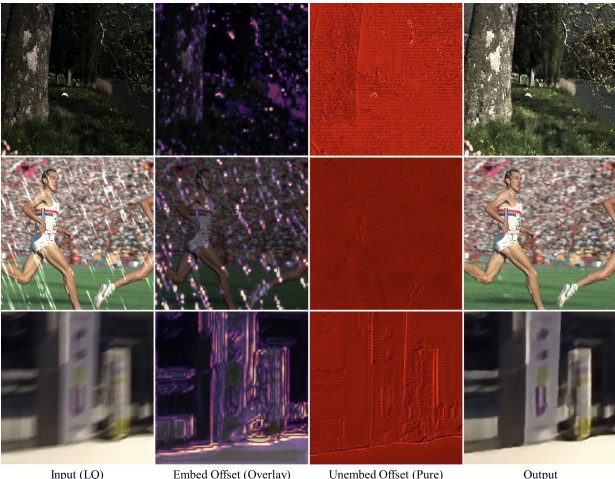

Input (LQ)     Embed Offset (Overlay)     Unembed Offset (Pure)     Output

*Figure 8.* Visualization of deformable tokenization offsets across representative restoration tasks. For each example (row), we show the degraded input (LQ), the embed-stage offset magnitude overlaid on the input, the unembed-stage offset magnitude map, and the restored output, illustrating how offsets respond to degradation patterns and structural regions.

### 4.4. Analysis and Visualization

**Grid Artifact Analysis.** Figure 7 analyzes the grid artifact bottleneck. We quantify boundary artifacts with a patch-boundary Grid Score:

$$\mathrm{GS}(\hat{\mathbf{y}}) = \frac{1}{2} \left( \mathbb{E}_{p \in \mathcal{B}_v} \|\Delta_x \hat{\mathbf{y}}(p)\|_1 + \mathbb{E}_{p \in \mathcal{B}_h} \|\Delta_y \hat{\mathbf{y}}(p)\|_1 \right),$$
(9)

where $\mathcal{B}_v = \{(i,j) \mid j \bmod P = 0\}$ and $\mathcal{B}_h = \{(i,j) \mid i \bmod P = 0\}$ denote vertical and horizontal patch-boundary pixels for patch size $P$, and $\Delta_x, \Delta_y$ are finite differences normal to the corresponding boundary directions. Lower values indicate weaker seam-like discontinuities; Table 2 reports this metric. As degradation severity increases (top row), JIT's grid artifact score rises sharply while restoration quality degrades. In contrast, FIT with deformable tokenization maintains stable artifact levels and achieves consistent improvements across all degradation levels. When scaling model capacity (bottom row), JIT plateaus early due to grid artifacts becoming the performance bottleneck, whereas FIT continues to benefit from increased capacity, achieving up to +2.4 dB improvement over JIT at larger model sizes.

**Learned Offset Visualization.** Figure 8 visualizes the learned sampling offsets across different restoration tasks. For each example, we show the degraded input, embed-stage offset magnitude overlaid on the input, unembed-stage offset magnitude map, and the restored output. Several interpretable patterns emerge: (1) On rainy images, offsets are larger along rain streak directions, suggesting the network learns to sample across streaks rather than along them; (2) On hazy images, offset magnitudes correlate with scene depth, larger offsets in distant (more degraded) regions; (3)

On blurry images, offsets concentrate around edges and high-frequency regions where blur is most visually apparent. This emergent behavior provides interpretable evidence that DGDT learns to adapt tokenization to degradation structure without explicit supervision on offset patterns.

**Why JIT as the Baseline?** JIT directly predicts clean images rather than noise or residuals, which intuitively aligns with the restoration objective and makes it a strong baseline for unified restoration. However, this formulation amplifies the impact of fixed patch boundaries: clean images exhibit strong spatial coherence, so neighboring patch regions must agree in color, texture, and structure. Any mismatch across a fixed grid therefore becomes visually prominent, especially when degradations vary spatially. In contrast, noise-predicting models are less affected since noise lacks semantic continuity. This is reflected in Table 3, where JIT shows a larger performance drop in multi-task settings compared to other methods. By making patch boundaries adaptive to degradation structure, FIT preserves the semantic coherence that direct prediction requires while eliminating the grid artifact bottleneck (Fig. 2).

## 5. Conclusion

We presented FIT, a unified image restoration framework that makes the entire pipeline degradation-aware through deformable tokenization. The core insight is that patch boundaries and reconstruction should adapt to local degradation severity, not remain fixed. By predicting global vector $\mathbf{g}$ and spatial map $\mathbf{M}$ to guide FiLM-modulated deformable embedding and unembedding, FIT effectively mitigates grid artifacts that bottleneck prior fixed-patch approaches. Combined with task-token dropout for robust degradation inference, FIT achieves SOTA results on five restoration tasks. More broadly, our results suggest that tokenization geometry is an active component of restoration quality: adapting the image-token interface itself can improve both quantitative accuracy and visible boundary consistency under heterogeneous degradations.

## Acknowledgements

This research is partially supported by the Ministry of Education, Singapore, under the Academic Research Fund Tier 1 (FY2026).

## Impact Statement

FIT may benefit archival preservation, visual communication, and downstream perception under adverse imaging conditions. However, restoration models can alter visual evidence or fail under domain shift, especially in medicine, forensics, and autonomous perception. High-stakes applications should validate FIT on target data, report failure cases, and not treat outputs as expert-reviewed evidence.

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

## Appendix

### Limitations

FIT trades the simplicity of fixed tokenization for a more expensive image-token interface. The overhead does not come from the Transformer blocks themselves, but from operations around them: predicting offsets, bilinear sampling during patch embedding, and inverse deformation during reconstruction. In our implementation this gives a 15–20% slowdown relative to JIT with comparable backbone capacity, and the gap is more visible in wall-clock latency than in FLOPs because irregular sampling is less hardware-friendly than dense matrix operations. This cost is justified when fixed grids cut across structured degradations, such as rain streaks, depth-varying haze, motion-blurred edges, or spatially uneven low-light noise. For nearly homogeneous corruptions, however, the benefit of deforming every patch boundary may be smaller. The current model learns small offsets in easier regions but does not impose an explicit budget on where deformation is allowed, so real-time deployments would likely need a gated version that applies deformation only in high-uncertainty or high-degradation regions. The sensitivity results in Table 7 also suggest that the chosen patch size and offset bound are robust operating points, not universally optimal constants.

**Evaluation Boundary.** The experiments are designed for controlled comparison across standard restoration benchmarks. The mixed-degradation study in Table 6 and the zero-shot results in Table 8 broaden this setting, but they still cover only a small subset of real imaging pipelines. Real photographs may contain a camera ISP, compression history, rolling shutter artifacts, unknown blur kernels, and sensor noise that do not match the training protocol. In settings where restored images are used for downstream decisions, especially autonomous perception or medical imaging, FIT should be validated on the target data distribution and inspected through failure cases in addition to aggregate PSNR/SSIM.

### Additional Protocol and Metric Details

**Mixed-Degradation Protocol.** The mixed-degradation control study in Table 6 is designed to test whether FIT learns degradation-aware local structure rather than only generic image content. We restrict the study to three reliably synthesized degradations on held-out clean natural images: Gaussian noise, motion blur, and low-light. For each image, two degradation types are sampled. Given a clean image $\mathbf{y}$, two degradation operators $\mathcal{D}_a$ and $\mathcal{D}_b$, and a smooth random mask $\mathbf{m} \in [0,1]^{H \times W}$, the mixed input is constructed as

$$\tilde{\mathbf{x}} = \mathbf{m} \odot \mathcal{D}_a(\mathbf{y}; \theta_a) + (1 - \mathbf{m}) \odot \mathcal{D}_b(\mathbf{y}; \theta_b), \tag{10}$$

where $\theta_a$ and $\theta_b$ denote the sampled degradation parameters. The mask is used only for evaluation, not as model input. We define dominant, boundary, and light/non-dominant regions by

$$\Omega_a = \{p : m_p > \tau\}, \quad \Omega_\partial = \{p : \tau \le m_p \le 1 - \tau\}, \quad \Omega_b = \{p : m_p < 1 - \tau\}, \tag{11}$$

with $\tau = 0.4$ in our evaluation. For any region $\Omega$, the region-wise PSNR is computed as

$$\mathrm{PSNR}_\Omega = 10 \log_{10} \frac{1}{|\Omega|^{-1} \sum_{p \in \Omega} \|\hat{\mathbf{y}}_p - \mathbf{y}_p\|_2^2}. \tag{12}$$

This protocol allows us to test whether gains concentrate in degradation-dominant and boundary zones, rather than only reflecting generic image-detail modeling. At inference time, all compared variants receive only the degraded image; no oracle mixed label, region label, or region mask is provided.

**Grid Score Computation.** Grid Score measures seam-like discontinuities on patch boundaries. In implementation, we evaluate adjacent pixel pairs crossing vertical and horizontal patch boundaries:

$$\mathrm{GS}(\hat{\mathbf{y}}) = \frac{1}{2} \sum_{r \in \{v,h\}} \frac{1}{|\mathcal{B}_r|} \sum_{(p,q) \in \mathcal{B}_r} \|\hat{\mathbf{y}}_p - \hat{\mathbf{y}}_q\|_1, \tag{13}$$

where $\mathcal{B}_v$ and $\mathcal{B}_h$ denote adjacent pixel pairs straddling vertical and horizontal patch boundaries, respectively. Outer image borders are excluded. For color images, the $\ell_1$ difference is computed over RGB channels and then averaged over all boundary pairs. Lower Grid Score indicates weaker artificial boundary discontinuities. This metric is used for the ablation in Table 2 and for the boundary visualizations in Figure 10.

**Offset Bound and Patch-Size Sensitivity.** Table 7 expands the sensitivity study summarized in the main text. When varying $\delta_{\max}$, the patch size is fixed to $P = 16$; when varying $P$, the normalized offset bound follows the default $\delta_{\max} = 0.25P$. FIT is stable across a reasonable offset range, with the best trade-off at $\delta_{\max} = 0.25P$. Smaller offset bounds limit deformation flexibility, while larger bounds bring little additional gain. For patch size, $P = 8$ partly reduces the fixed-grid problem even without large deformations, whereas $P = 32$ aggravates grid artifacts; the best operating point is $P = 16$.

*Table 7.* Sensitivity to offset bound and patch size on the five-degradation setting. Values are average PSNR.

| Group | Setting | 5-task Avg. PSNR |
|---|---|---|
| Offset bound | $\delta_{\max} = 0.125P$ | 30.41 |
| Offset bound | $\delta_{\max} = 0.25P$ | **30.72** |
| Offset bound | $\delta_{\max} = 0.375P$ | 30.70 |
| Patch size | $P = 8$ | 30.66 |
| Patch size | $P = 16$ | **30.72** |
| Patch size | $P = 32$ | 30.21 |

**Task-Token Dropout Inference Protocol.** Table 5 evaluates task-token robustness under four inference settings. In the hard-token setting, the correct task token is provided, corresponding to the standard case with known degradation type. In the soft-token setting, no explicit task label is used; tokens are generated from the degradation representation **g**. In the wrong-token setting, each test image with ground-truth label $k$ is evaluated with the remaining $K - 1$ incorrect task tokens, and PSNR is averaged over all mismatched assignments to measure sensitivity to unreliable metadata. In the mixed-degradation setting, inputs follow the mixed protocol above and no unique hard task label is assumed. TTD is applied only during training: with probability $p_{\mathrm{drop}}$, the hard token is replaced by the soft token derived from **g**. Thus, at test time, FIT can use either explicit task tokens when available or image-derived soft tokens when the task identity is missing, ambiguous, or spatially mixed.

**Additional Generalization Study**

**Zero-Shot Unseen Degradations.** We also evaluate whether the learned degradation encoder transfers beyond the five degradation types used for training. Without fine-tuning, the same checkpoints are tested on three unseen degradations: JPEG compression artifacts, real-world noise, and defocus blur. Table 8 shows consistent gains over JIT, suggesting that the image-derived degradation representation captures transferable corruption structure rather than only memorizing training labels.

*Table 8.* Zero-shot generalization to unseen degradations. The model is trained on the five-degradation setting and evaluated without fine-tuning. Values are PSNR.

| Unseen Degradation | Dataset | JIT | FIT |
|---|---|---|---|
| JPEG artifacts (QF=10) | LIVE1 | 27.63 | **28.11** |
| Real-world noise | SIDD(val) | 29.21 | **29.78** |
| Defocus blur | DPDD(indoor) | 23.41 | **23.94** |

**Discussion of the Unseen-Degradation Results.** Although the three degradations above are absent from training, FIT improves over JIT by 0.48–0.57 dB on all of them, indicating that the learned offsets generalize to corruption structures (block artifacts, real sensor noise, optical defocus) outside the rain/haze/blur/Gaussian-noise/low-light training mix. The largest absolute gap, +0.57 dB on SIDD real-world noise, is consistent with our hypothesis that degradation-aware tokenization helps most when the corruption is spatially heterogeneous: real sensor noise correlates with local intensity and texture, whereas the Gaussian noise seen during training is approximately stationary, so the offset prior has room to adapt. On JPEG artifacts the gain (+0.48 dB) is the smallest, because $8 \times 8$ blocking is itself a fixed-grid degradation; FIT can still reduce seam visibility but the deformation prior is less informative than for streak- or kernel-like structures. At inference, no task token or oracle label is provided—the soft token derived from **g** is used in all three rows, mirroring the soft-token protocol of Table 5.

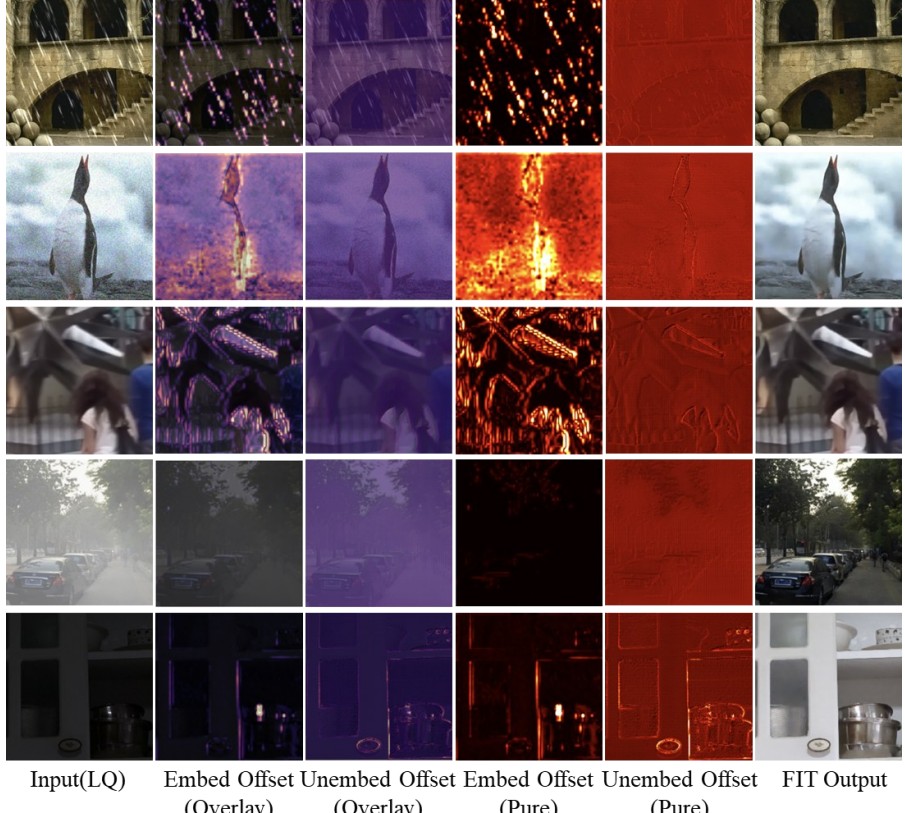

| Input(LQ) | Embed Offset (Overlay) | Unembed Offset (Overlay) | Embed Offset (Pure) | Unembed Offset (Pure) | FIT Output |

*Figure 9.* Visualization of learned offsets across different degradation tasks. From top to bottom: deraining, deraining, deblurring, dehazing, and low-light enhancement. Embed offsets respond to degradation-specific structures (e.g., rain streaks and motion edges), while unembed offsets remain smoother for reconstruction.

## Additional Visualization and Analysis

This section complements the quantitative results with qualitative visualizations of the learned deformable tokenization, including per-task offset patterns (Figure 9), patch-boundary artifact maps (Figure 10), and FIT outputs on composite degradations (Figure 11).

**Detailed Offset Behavior Analysis.** Figure 9 provides extended visualization of the learned deformable tokenization offsets across five representative restoration tasks. For each example, we display six views: the degraded input (LQ), embed-stage offset magnitude as an overlay on the input, embed-stage offset magnitude as a pure heatmap, unembed-stage offset magnitude as an overlay, unembed-stage offset magnitude as a pure heatmap, and the final restored output.

Several task-specific patterns emerge from this visualization. For deraining (Rows 1-2), embed offsets exhibit strong responses along rain streak orientations, suggesting the network learns to sample perpendicular to streaks, effectively "looking through" the rain. For deblurring (Row 3), offset magnitudes concentrate around edges and high-frequency regions where motion blur is most perceptually significant, while smooth regions receive smaller offsets. For dehazing (Row 4), offset magnitude correlates with scene depth—distant hazy regions exhibit larger offsets than nearby clear regions, demonstrating that the degradation encoder captures the spatial structure of atmospheric scattering without explicit depth supervision. For low-light enhancement (Row 5), offsets are larger in extremely dark regions where noise and degradation are most severe, helping the model focus its deformation capacity where most needed.

**Grid Artifacts and Composite Degradation Examples.** Figures 10 and 11 provide additional visual evidence for two challenging settings. Figure 10 compares JIT and FIT with patch-boundary gradient maps and Grid Scores, while Figure 11 shows qualitative FIT results on CDD-11 composite degradations.

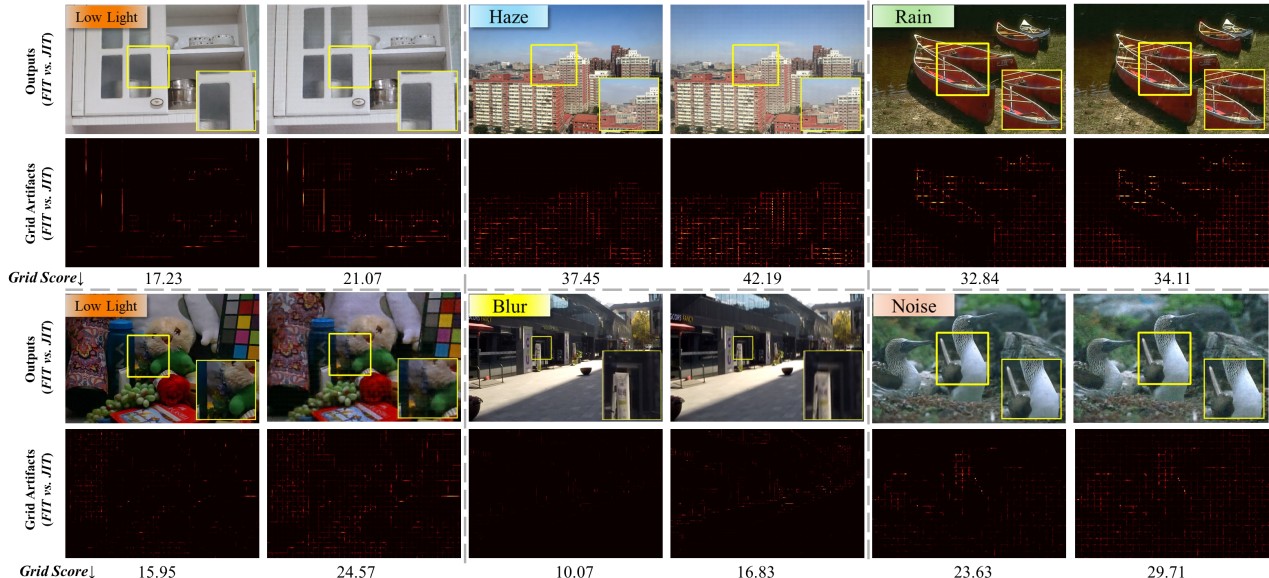

*Figure 10.* Additional grid artifact visualizations. We compare FIT and JIT across representative restoration tasks using restored outputs, patch-boundary artifact maps, and Grid Scores. Lower Grid Score indicates weaker grid-like discontinuities.

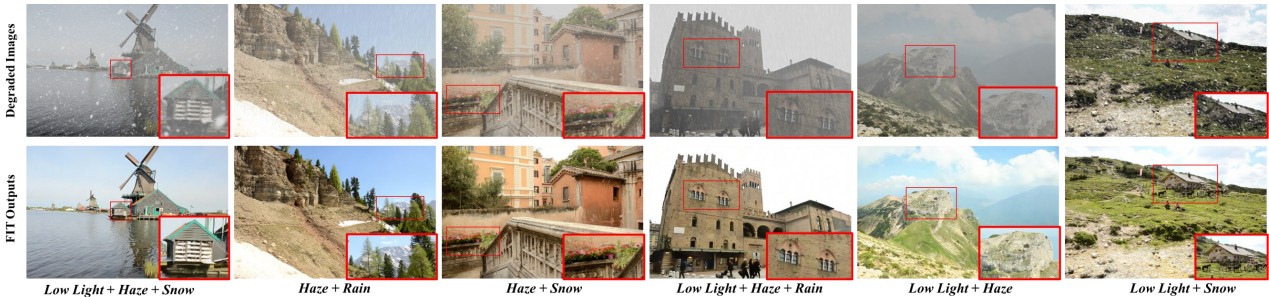

*Figure 11.* Additional composite-degradation examples. The first row shows degraded inputs with mixed degradation types and zoomed regions, and the second row shows FIT outputs on CDD-11 composite degradations.

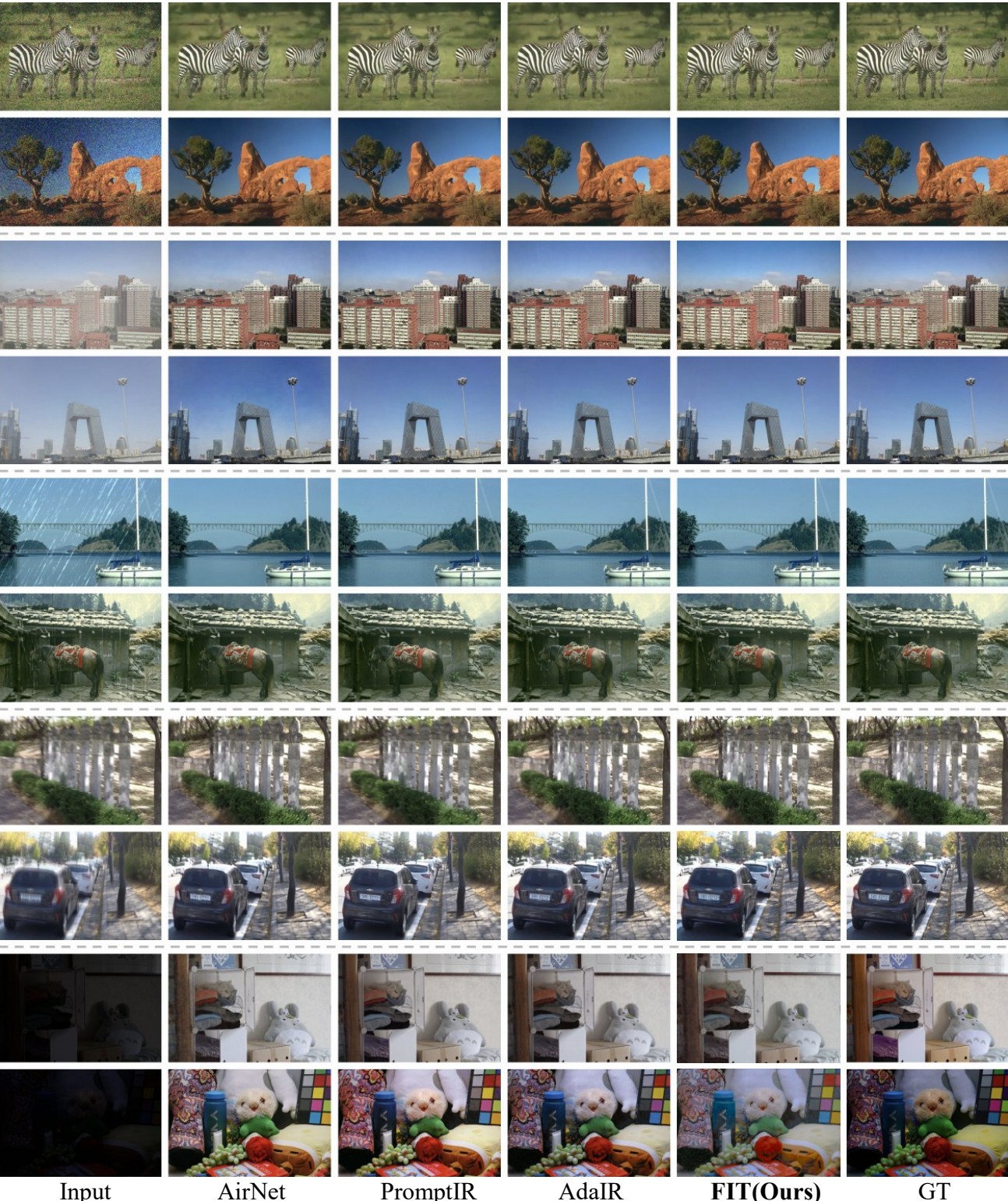

| Input | AirNet | PromptIR | AdaIR | **FIT(Ours)** | GT |

*Figure 12.* Visual comparisons of FIT against state-of-the-art All-in-One methods under Single Degradation task on BSD68, SOTS-Outdoor, and Rain100L datasets (from top to bottom). Zoom-in for best view.

