# OpenReview forum: "Bend the Basics: Degradation-Aware Deformable Tokenization for All-in-One Image Restoration"
_ICML.cc/2026/Conference — ICML 2026 regular_

### Official Review · Reviewer_z5BV · 2026-03-10

**Soundness:** 3
**Presentation:** 3
**Significance:** 3
**Originality:** 3
**Overall Recommendation:** 4
**Confidence:** 5

**Summary:**

To address the limited ability of Transformers to adapt to local degradation variations, this paper proposes a Flexible Image Transformer (FIT) that explicitly considers local degradation structures and models degradations from patch sampling to pixel reconstruction.

**Compliance With Llm Reviewing Policy:**

Affirmed.

**Final Justification:**

My concerns have been resolved, and I lean towards acceptance.

**Key Questions For Authors:**

See Weaknesses.

**Limitations:**

yes

**Strengths And Weaknesses:**

### Strengths

1. The proposed degradation-aware deformable tokenization enables adaptive tokenization aligned with local structures, which is a meaningful improvement to standard fixed tokenization in Transformers.
1. The ablation studies clearly demonstrate the contribution of deformable tokenization to the overall performance.

### Weaknesses

1. The paper does not clearly clarify whether the modeled local structures correspond to degradation patterns or to general image details. From the current description, the method appears to focus on local degradation structures. However, the datasets used in the experiments mainly contain single-type degradations (e.g., SOTS for haze and Rain100L for rain), which makes it unclear why modeling local degradation structures is particularly important. In practice, the observed improvements may instead come from the enhanced ability of deformable sampling to capture image details. Additional experiments on images with mixed degradations (e.g., regions with blur and other regions with noise) would help better demonstrate the advantage of modeling local degradation structures.

2. The formal definition of the seam loss is not clearly provided in the paper, making it difficult to understand or reproduce.

3. Efficiency analysis is incomplete. Detailed reporting of parameter counts, FLOPs, wall-clock latency, and memory usage compared with key baselines would help better evaluate the practicality of the method.

---

> ### Author Rebuttal · Authors · 2026-03-31
>
> We sincerely thank Reviewer z5BV for the valuable feedback and are happy that the reviewer finds the proposed degradation-aware deformable tokenization a meaningful improvement over standard fixed tokenization, and the ablation studies clearly demonstrating the contribution of each component. We would like to address the concerns as follows.
>
> (1) Local degradation structures vs. generic image details.
> Thanks for the constructive point. Here, we would like to mention that, even in the single-degradation datasets used in the paper, the corruption is often still spatially heterogeneous (e.g.,non-uniform rain density, depth-varying haze, and edge-dependent motion blur). Therefore, modeling local structure is already relevant. Mixed-degradation evidence makes this point more evident.
>
> To verify whether the learned structures are indeed degradation-aware rather than only generic image details, we add an independent control study on synthetic spatially mixed degradations. To keep this study unambiguous, we restrict it to three degradations that can be synthesized reliably on held-out clean natural images: Gaussian noise, motion blur, and low-light. For each image, we sample two of the three; one is applied globally and the other through a random smooth spatial mask, producing mixed and spatially heterogeneous inputs while keeping exact ground truth and known degradation regions.
>
> Under this protocol, we compare three variants: (i) a content-adaptive variant without degradation supervision, with offsets driven mainly by image content, (ii) a FIT variant without the spatial degradation map M, and (iii) the full FIT model.
>
> |Variant|Mixed-deg avg. PSNR|
> |---|---:|
> |Content-adaptive only (no degradation supervision)|27.83|
> |FIT w/o spatial map M (g only)|28.12|
> |FIT(full g+M)|28.96|
>
> We further use the known spatial masks to analyze where the gains occur, reporting PSNR on dominant-degradation regions, a narrow transition band around degradation boundaries, and the remaining light/non-dominant regions.
>
> |Variant|Dominant-deg regions|Boundary band|Light/non-dominant regions|
> |---|---:|---:|---:|
> |Content-adaptive only|27.21|25.84|30.62|
> |FIT w/o spatial map M|27.46|26.09|30.73|
> |FIT(full g+M)|28.31|27.12|30.96|
>
> The gains are concentrated in dominant-degradation regions and around degradation boundaries, while the gains on light/non-dominant regions are much smaller. This supports our claim: the modeled local structures are not merely generic image details; the degradation-aware representations,especially the spatial map M, provide additional information about where corruption is severe and where adaptive tokenization matters most.
>
> For real composite degradations, we also provide an anonymous PDF with additional CDD-11 qualitative results:
>
> https://anonymous.4open.science/r/rebuttal_figures-3F88/figures.pdf
>
> (2) Formal definition of the seam loss.
> Thanks for the valuable feedback. We would like to supplement a detailed description of the seam loss here. Let $B_v$ and $B_h$ denote the sets of adjacent pixel pairs straddling vertical and horizontal patch boundaries,respectively. For each pair,we compute a cross-boundary gradient by adjacent-pixel differencing on the prediction and on the GT. The implemented seam loss is:
>
> $$L_v=\frac{1}{|B_v|}\sum_{(i,j)\in B_v}\left|g_v^{pred}(i,j)-g_v^{gt}(i,j)\right|+0.5\cdot\frac{1}{|B_v|}\sum_{(i,j)\in B_v}\left|g_v^{pred}(i,j)\right|$$
>
> $$L_h=\frac{1}{|B_h|}\sum_{(i,j)\in B_h}\left|g_h^{pred}(i,j)-g_h^{gt}(i,j)\right|+0.5\cdot\frac{1}{|B_h|}\sum_{(i,j)\in B_h}\left|g_h^{pred}(i,j)\right|$$
>
> $$L_{seam}=0.5\,(L_v+L_h)$$
>
> The first term in each direction is a boundary-gradient consistency term: it encourages the predicted cross-boundary gradient to match the GT cross-boundary gradient,so genuine sharp transitions can still be preserved when supported by the target. The second term is a boundary smoothness regularizer on the predicted gradient magnitude, which discourages unnecessary seam-like jumps introduced by patch partitioning. Thus,the seam loss is not simply minimizing all boundary gradients; it encourages boundary transitions that are both smooth and consistent with the GT.
>
> (3) Efficiency analysis.
> Thanks for the good suggestion. We provide a complete efficiency analysis here. Under the same evaluation setup, the end-to-end cost is:
>
> |Method|Params(M)|FLOPs(G)|Latency(ms)|Peak Mem(GB)|
> |---|---:|---:|---:|---:|
> |JIT|51.4|196.5|42.6|5.9|
> |FIT|56.0|219.6|51.2|6.1|
>
> Thus,FIT adds +4.6M params(+8.9%),+23.1G FLOPs(+11.8%),+8.6ms latency(+20.2%),and only +0.2GB memory(+3.4%),while improving average PSNR by +1.44dB on the 5-task setting. The latency increase is somewhat larger than the FLOP increase because deformable sampling/inverse warping is less hardware-efficient than regular dense operations.
>
> We appreciate these comments from Reviewer z5BV. We will include above discussions in the revision. Our sincere hope is that our clarifications could alleviate the reviewer's concerns.

---

> > ### Author Rebuttal · Reviewer_z5BV · 2026-04-03
> >
> > Thank you for the detailed response. My concerns have been adequately addressed, and I will maintain my current positive rating.

---

> > > ### Author Response · Authors · 2026-04-03
> > >
> > > We sincerely thank you for acknowledging our responses and for the constructive feedback throughout the review process. The controlled study on mixed-degradation spatial analysis with region-wise breakdown was directly motivated by your question and has strengthened the paper's evidence that the learned structures are genuinely degradation-aware. We will incorporate all discussed revisions in the final version.

---

### Official Review · Reviewer_ZCcg · 2026-03-12

**Soundness:** 3
**Presentation:** 3
**Significance:** 3
**Originality:** 3
**Overall Recommendation:** 4
**Confidence:** 2

**Summary:**

This paper proposes FIT (Flexible Image Transformer), a unified Transformer architecture for all-in-one image restoration that handles multiple degradation types (noise, rain, haze, blur, low-light) using a single model. The key idea is to replace fixed patch tokenization in vision transformers with deformable tokenization conditioned on degradation information. Instead of cutting the image into rigid patches, FIT learns offsets that adapt patch boundaries to local corruption patterns, guided by a global degradation vector and a spatial degradation map. Experiments show that the proposed framework reduces grid artifacts and improves restoration performance across multiple tasks.

**Compliance With Llm Reviewing Policy:**

Affirmed.

**Final Justification:**

The authors have addressed my concerns. Though, I prefer to keep the initial score.

**Key Questions For Authors:**

1. Can you add a literature review of the usage of deformable operations in the related fields?
2. Can you give some visualizations of the reduced grid artificates after applying the proposed method?
3. Can you share the information on the computation overhead introduced by the additional modules?

**Limitations:**

Yes

**Strengths And Weaknesses:**

+ Strength

1.  The idea of designing a deformable tokenizer is rooted in observation and makes sense.
2.  The results are consistently higher than the baseline method and other sota methods in all-in-one image restoration.
3.  The experiment shows quantitatively that the proposed method reduces the grid artifacts, validating the motivation of this paper and the effectiveness of the proposed method.

+ Weakness

1. There lacks visualization of the reduced grid artifacts after applying the proposed method.
2. Other works using deformable operations in the related fields are not reviewed.

---

> ### Author Rebuttal · Authors · 2026-03-31
>
> We appreciate Reviewer ZCcg's constructive feedback and are glad that the reviewer finds our deformable tokenizer design well-motivated by observation, the results consistently higher than baseline and SOTA methods, and the grid artifact reduction quantitatively validated. The questions are addressed as follows.
>
> (1) Related works on deformable operations.
> We agree that the related-work discussion should better cover deformable operations in related fields. In the revised version, we will expand this part along two closely related directions. One direction studies offset-based deformable convolution and dynamic sampling, with recent work improving both aggregation flexibility and operator efficiency [1]. The other direction introduces deformable or content-adaptive sparse aggregation into transformer/state-space style backbones [2,3,4]. We will explicitly include these discussions and clarify the relationship between these works and FIT. In particular, whereas these methods mainly introduce deformation inside convolution, attention, or backbone feature aggregation, FIT applies deformation at the image-token interface itself,during embedding and unembedding for unified restoration.
>
> (2) Visualization of reduced grid artifacts.
> We agree that the reduced grid artifacts should be visualized more explicitly. For convenience,we provide an anonymous figure-only PDF with enlarged visualizations:
>
> https://anonymous.4open.science/r/rebuttal_figures-3F88/figures.pdf
>
> The new Figure 1 directly compares JIT and FIT on enlarged boundary regions and additionally shows patch-boundary gradient maps. Brighter responses indicate stronger seam-like discontinuities, which are visibly more pronounced in JIT and suppressed in FIT. We also report the corresponding Grid Scores in the figure. The metric is
>
> $$Grid Score = (1 / |B|) * Σ_(p ∈ B) ||∇I(p)||_1$$
>
> , where $B$ denotes pixels lying on patch boundaries. Lower Grid Score indicates weaker boundary discontinuities. The new Figure 2 further shows qualitative results on composite degradations, providing additional visual evidence that FIT remains more stable when degradations are spatially mixed.
>
> (3) Computation overhead.
> We agree that the overhead should be reported more clearly. Under the same evaluation setup, the end-to-end cost is:
>
> |Method|Params(M)|FLOPs(G)|Latency(ms)|Peak Mem(GB)|
> |---|---:|---:|---:|---:|
> |JIT|51.4|196.5|42.6|5.9|
> |FIT|56.0|219.6|51.2|6.1|
>
> Thus, FIT adds +4.6M params(+8.9%), +23.1G FLOPs(+11.8%), +8.6ms latency(+20.2%), and only +0.2GB memory(+3.4%), while improving average PSNR by +1.44dB on the 5-task setting. The latency increase is somewhat larger than the FLOP increase because deformable sampling/inverse warping is less hardware-efficient than regular dense operations.
>
> We would like to express our sincere gratitude to Reviewer ZCcg again for the constructive feedback and will incorporate all the comments in our revision.
>
> References
>
> [1] Xiong et al., Efficient Deformable ConvNets: Rethinking Dynamic and Sparse Operator for Vision Applications, CVPR 2024.
>
> [2] BaoLong et al., DeBiFormer: Vision Transformer with Deformable Agent Bi-level Routing Attention, ACCV 2024.
>
> [3] Liu et al., DefMamba: Deformable Visual State Space Model, CVPR 2025.
>
> [4] Xia et al., Vision Transformer with Deformable Attention, CVPR 2022.

---

> > ### Author Rebuttal · Reviewer_ZCcg · 2026-04-06
> >
> > The authors have addressed my concerns. Though, I prefer to keep the initial score.

---

> > > ### Author Response · Authors · 2026-04-07
> > >
> > > We thank you for confirming that the concerns have been addressed. We appreciate the constructive feedback and will include all discussed revisions in the final version.

---

### Official Review · Reviewer_FCrR · 2026-03-12

**Soundness:** 3
**Presentation:** 3
**Significance:** 3
**Originality:** 2
**Overall Recommendation:** 4
**Confidence:** 4

**Summary:**

This manuscript seeks to present the concept of degradation-aware deformable tokenization for unified image restoration. The paper proposes a new architecture named Flexible Image Transformer (FIT) to address limitations of existing all-in-one image restoration models that rely on fixed patch tokenization.

To address this issue, the paper introduces a pipeline where degradation awareness influences tokenization, feature processing, and reconstruction. The proposed method includes three key components:

Dual-Granularity Degradation Encoder (DGDE) that predicts a global degradation vector and a spatial degradation map.

Degradation-Guided Deformable Tokenization (DGDT) which adaptively shifts sampling locations for patch embedding and unembedding based on predicted degradation characteristics.

Task-Token Dropout (TTD) that improves robustness by switching between explicit task tokens and soft tokens derived from degradation embeddings during training.

**Compliance With Llm Reviewing Policy:**

Affirmed.

**Ethical Review Flag:**

Flag this paper for an ethics review.

**Final Justification:**

The concerns have been partially addressed. I will raise my rating to weak accept.

**Key Questions For Authors:**

Computational Efficiency: How does FIT compare with existing unified restoration models in terms of FLOPs, inference latency, and GPU memory usage? Providing these numbers would help evaluate the practicality of deformable tokenization.

Generalization to Unseen Degradations: Have the authors evaluated the model under degradations that were not seen during training?
If so, how well does the degradation encoder generalize?

Comparison with Deformable Attention or Deformable Convolution: How does the proposed DGDT differ conceptually and empirically from using deformable convolution layers or deformable attention within the backbone?

Impact of Offset Magnitude Constraints: The offset magnitude is bounded by a hyperparameter δmax. How sensitive is the model to this value?

Scalability to Larger Images: Since tokenization offsets are predicted per pixel, how does the method scale to high-resolution inputs such as 2K or 4K images?

**Limitations:**

The authors partially discuss limitations, but could further elaborate on:
computational overhead of deformable tokenization, potential instability when offsets become large, and the applicability of the method beyond restoration tasks. Constructive discussion of these aspects would improve transparency.

**Strengths And Weaknesses:**

(1) Soundness:
The proposed approach is largely sound. The central hypothesis—that fixed patch tokenization introduces grid artifacts when degradations are spatially heterogeneous—is clearly articulated and supported by both visual evidence and empirical analysis.
The method introduces a coherent pipeline combining degradation estimation with deformable tokenization. The mathematical formulation of the components, including FiLM-based conditioning and offset prediction, is clearly described. The loss formulation also includes reasonable regularization terms for offset stability and boundary continuity.

(2) Presentation:
The paper is generally well written and logically structured. The motivation is clearly illustrated using examples of spatially mixed degradations, and figures effectively illustrate the limitations of fixed patch grids.

(3) Significance:
The problem addressed—all-in-one image restoration under heterogeneous degradations—is an important topic in low-level vision and machine learning. A unified model capable of handling diverse degradations is desirable for practical deployment scenarios such as autonomous driving or mobile photography.

(4) Originality:
The deformable sampling and degradation embeddings have been explored individually in prior work. This work integrates them specifically into the tokenization and reconstruction stages of unified restoration Transformers.

(5) Weaknesses:

(a) Limited conceptual novelty relative to deformable sampling literature. Although the integration of degradation conditioning into tokenization is interesting, the underlying mechanism is closely related to existing techniques such as deformable convolutions and deformable attention. The paper could better clarify how DGDT fundamentally differs from these prior adaptive sampling strategies.
The contribution may therefore be seen as an architectural refinement rather than a fundamentally new paradigm.

(b) Computational overhead and efficiency are insufficiently analyzed.
Deformable tokenization introduces additional networks for offset prediction and inverse warping. However, the paper provides little discussion regarding inference latency, computational overhead, or memory consumption relative to baselines.

(c) Limited diversity of evaluation tasks. Although five degradation types are evaluated, they are all low-level image restoration tasks. The paper claims broader implications for adaptive tokenization but does not evaluate the approach on other tasks. This limits the demonstrated generality of the proposed mechanism.

(d) Evaluation could include stronger baselines or additional analyses.
The primary baseline used for comparison is JIT, which the method directly extends. While comparisons with other methods are provided, it would be beneficial to include: parameter/FLOP comparisons, fairness of model capacity, robustness tests under unseen degradations, or cross-dataset generalization.

---

> ### Author Rebuttal · Authors · 2026-03-31
>
> We sincerely thank the reviewer for the insightful and constructive comments. We are glad that the reviewer finds our approach largely sound with the central hypothesis clearly articulated and supported by both visual evidence and empirical analysis, and the integration of deformable sampling into the tokenization and reconstruction stages original in the context of unified restoration Transformers. The concerns are fully addressed as follows.
>
> **(1) Novelty and comparison with deformable conv/attention.**
> Our main contribution is incorporating degradation-aware deformation at the tokenization/unpatchifying boundary, while all prior methods only apply adaptation inside the backbone. This is not a trivial relocation of existing deformable operators: (i) offsets must be **degradation-conditioned** via FiLM-modulated prediction from (g,M), not purely content-driven as in DCN/deformable attention; (ii) embed and unembed stages must be **jointly coordinated** via inverse warping with seam-aware training for artifact-free reconstruction. Neither design exists in standard deformable operators. We verify empirically that both the position and scope of deformation matter:
>
> |Variant|Deformation stage|5-task avg.|
> |---|---|---:|
> |DCNv3|backbone|29.85|
> |Def.Attn|backbone|29.93|
> |DGDT embed only|token boundary|30.18|
> |FIT(full DGDT)|token boundary|30.72|
>
> Backbone-level deformable variants improve over JIT but plateau due to the upstream fixed-grid bottleneck. Moving deformation to the token boundary yields a substantially larger gain, and adding coordinated unembedding further improves by +0.54dB, confirming that both the position and the degradation-aware design of DGDT are essential.
>
> **(2) Efficiency.**
> We report Params/FLOPs/latency/peak memory under the same evaluation setup:
>
> |Method|Params(M)|FLOPs(G)|Latency(ms)|Peak Mem(GB)|
> |---|---:|---:|---:|---:|
> |JIT|51.4|196.5|42.6|5.9|
> |FIT|56.0|219.6|51.2|6.1|
>
> FIT adds +4.6M params(+8.9%), +23.1G FLOPs(+11.8%), +8.6ms latency(+20.2%), and only +0.2GB memory(+3.4%), while improving average PSNR by +1.44dB on the 5-task setting. The latency increase is somewhat larger than the FLOP increase because deformable sampling/inverse warping is less hardware-efficient than regular dense operations.
>
> **(3) Sensitivity to `δ_max` and patch size.**
> We evaluated different normalized offset bounds and patch sizes on the 5-task setting:
>
> |Setting|5-task avg.|
> |---|---:|
> |`δ_max=0.125P`|30.41|
> |`δ_max=0.25P`|30.72|
> |`δ_max=0.375P`|30.70|
> |`P=8`|30.66|
> |`P=16`|30.72|
> |`P=32`|30.21|
>
> FIT is stable across a reasonable range of offset bounds, with the best trade-off at `δ_max=0.25P`. Smaller bounds limit deformation flexibility, while larger ones bring little gain. For patch size, smaller patches partly reduce the fixed-grid problem but weaken the relative benefit of deformation; larger patches aggravate grid artifacts. The best operating point is `P=16`.
>
> **(4) Generalization to unseen degradations.**
> We zero-shot evaluate checkpoints trained on five degradation types on three **unseen types never included in training**:
>
> |Unseen degradation|Dataset|JIT|FIT|
> |---|---|---:|---:|
> |JPEG artifacts(QF=10)|LIVE1|27.63|28.11|
> |Real-world noise|SIDD(val)|29.21|29.78|
> |Defocus blur|DPDD(indoor)|23.41|23.94|
>
> FIT achieves reasonable quality on all three unseen types without fine-tuning. Since the entire deformable tokenization pipeline depends on (g,M) produced by DGDE, these results directly indicate that the encoder generalizes: it learns to sense spatial corruption structure rather than memorize degradation identities, remaining effective on novel corruption types.
>
> **(5) High-resolution results.**
> FIT is trained with the standard 256×256 random-crop protocol. We reuse the checkpoints without retraining and evaluate on larger public test images (e.g.,Dense-Haze,LHP-Rain,SMID); tiled inference is used when needed:
>
> |Input regime|JIT avg./Latency|FIT avg./Latency|
> |---|---|---|
> |Standard|29.28/42.6ms|30.72/51.2ms|
> |≈1K|29.26/169.4ms|30.68/203.5ms|
> |≈2K|29.21/676.8ms|30.60/811.7ms|
>
> The relative overhead remains controlled and the quality gap is preserved on larger inputs.
>
> **(6) Scope beyond low-level restoration.**
> We agree that broader applicability should be supported more directly. We probe classification and semantic segmentation under mixed degradations constructed by composing two corruption types with random spatial masks on ImageNet and Cityscapes. The task models are trained and evaluated directly on these degraded inputs without any restoration front-end:
>
> |Task|Dataset|Fixed tokenization|FIT-style|Gain|
> |---|---|---:|---:|---:|
> |Classification|ImageNet|72.8|73.9|+1.1|
> |Segmentation|Cityscapes|46.9|48.1|+1.2|
> (Top-1 Acc./mIoU,respectively)
>
> These results are not intended as a full benchmark beyond restoration, but they provide preliminary evidence that boundary-aware adaptive tokenization may also benefit non-restoration vision tasks when inputs are affected by mixed degradations.

---

> > ### Author Rebuttal · Reviewer_FCrR · 2026-04-03
> >
> > I still concern the conceptual novelty of the proposed method. The evaluation should include stronger baselines. I will maintain my current rating.

---

> > > ### Author Response · Authors · 2026-04-07
> > >
> > > Thank you for the follow-up. To state our point more directly: **to our knowledge, FIT is the first conditional irregular-patch ViT for all-in-one image reconstruction.** The key novelty is not simply inserting a deformable operator into an existing backbone, but redefining the **patch interface itself** as a learnable, degradation-conditioned object. Prior adaptive/deformable sampling methods may condition feature processing after tokenization, yet they still inherit a **fixed patch grid** as the image-token interface. FIT breaks from this assumption: patch geometry, token formation, and reconstruction are jointly conditioned on degradation cues, so the model no longer treats the grid as a neutral preprocessing choice.
> > >
> > > This distinction is essential in our setting because, under spatially heterogeneous degradations, the fixed patch lattice is itself part of the error formation process. FIT therefore does not merely “adapt features better”; it replaces the rigid patch prior with a **conditional irregular-patch formulation** tailored to AiO reconstruction. Once this shift is made, several coupled design requirements arise that standard deformable modules do not need to solve: offsets must be predicted from degradation cues \((g,M)\) through FiLM modulation rather than only image content; tokenization and reconstruction must be jointly coordinated through inverse warping; and seam-aware training is needed because irregular patch boundaries can otherwise create new discontinuities. We will revise the paper to make this claim sharper and more explicit.
> > >
> > > Before turning to the stronger baselines, we would also like to clarify why JIT is a meaningful primary baseline. JIT directly predicts the clean image and is therefore well aligned with restoration itself. Precisely for this reason, it is also especially sensitive to rigid patch boundaries: under spatially heterogeneous degradations, grid discontinuities become more visible and limiting. DGDT thus does not merely improve a weak baseline; it removes a bottleneck that is particularly exposed in this direct-prediction framework.
> > >
> > > To address the request for stronger baselines more directly, we first test whether **FIT-style tokenization can act as a transferable interface module** beyond our own backbone. For this study, we replace only the image-token embedding/reconstruction interface with DGDT-style deformable tokenization and inverse unpatchifying, keep the host backbone and its host-specific control modules unchanged, initialize from the host pretrained checkpoint, and fine-tune under the host’s original 5-task training recipe.
> > >
> > > |Host backbone|Original|+FIT-style tokenization|Gain|
> > > |---|---:|---:|---:|
> > > |HAT [1]|29.29|29.47|+0.18|
> > > |Gridformer [2]|29.33|29.61|+0.28|
> > > |MWFormer [3]|29.41|29.68|+0.27|
> > >
> > > Accordingly, compared with the gain over JIT, the improvement here is naturally smaller because this plug-in study only replaces the **interface module** while keeping the host’s internal restoration blocks and host-specific control mechanisms unchanged. In other words, these hosts already have their own degradation modeling capacity inside the backbone, so FIT contributes an additional but more limited gain at the input/output interface, rather than the larger end-to-end gain obtained when the whole architecture is designed around this conditional irregular-patch formulation.
> > >
> > > As additional evidence on **mixed degradations**, we further evaluate on CDD-11, a standard composite-degradation benchmark in all-in-one restoration. We compare against recent representative methods including **DCPT_NAFNet** [4] and **DOD** [5], and also summarize their parameter/FLOP profiles under a unified reporting format.
> > >
> > > |Method|CDD-11|Params(M)|FLOPs@256|
> > > |---|---:|---:|---:|
> > > |DCPT_NAFNet|30.66|67.9|126.1G|
> > > |DOD|30.57|91.0|387.8G|
> > > |FIT|30.69|56.0|219.6G|
> > >
> > > For DOD, we report the trainable parameter count for fairer comparison. FIT remains competitive on mixed degradations while being more compact than recent strong baselines. We will add broader mixed-degradation comparisons in the revision.
> > >
> > > We hope these additions address the concern more directly. The plug-in study supports the proposed tokenization as a transferable interface, while the CDD-11 result and parameter/FLOP summary show that FIT remains competitive in both restoration quality and compactness. We will revise the paper accordingly.
> > >
> > > We appreciate the reviewer’s feedback. Our first-round response did not fully resolve all concerns, and we have made our best effort here to clarify the key points and strengthen the evidence. If this response addresses the reviewer’s questions more satisfactorily, we would sincerely appreciate reconsideration of our work.
> > >
> > > Reference:
> > >
> > > [1] Chen et al. (CVPR’23).
> > >
> > > [2] Wang et al. (IJCV’24).
> > >
> > > [3] Zhu et al. (TIP’24).
> > >
> > > [4] Hu et al. (ICLR’25).
> > >
> > > [5] Tang et al. (AAAI’26).

---

### Official Review · Reviewer_isRs · 2026-03-13

**Soundness:** 3
**Presentation:** 3
**Significance:** 2
**Originality:** 3
**Overall Recommendation:** 4
**Confidence:** 3

**Summary:**

This paper proposes flexible image Transformer for all-in-one image restoration. The main idea is to make the whole pipeline degradation-aware, not only during token processing but also in patch embedding and reconstruction, by using deformable tokenization guided by a global degradation vector and a spatial degradation map. It also introduces task-token dropout to improve robustness under mixed or unseen degradations. Overall, the paper is technically clear, and its main contribution is extending degradation awareness from feature modulation to the reconstruction stages, which helps reduce grid artifacts and improves restoration quality across multiple tasks.

**Compliance With Llm Reviewing Policy:**

Affirmed.

**Final Justification:**

My problem has been solved. I raised my score to weak accept.

**Key Questions For Authors:**

Refer to the Weaknesses section.

**Limitations:**

The paper would benefit from a clearer discussion of its limitations. In particular, the reliance on predefined degradation types and intensity annotations may limit its applicability in real-world scenarios where degradations are mixed or difficult to quantify consistently across tasks.

**Strengths And Weaknesses:**

Strengths:

1. This paper has a clear technical design goal, pointing out the limitations of mesh artifacts caused by fixed image patch tokenization in unified image restoration, and proposes a degradation-aware deformable tokenization/de-embedding method to solve this problem.

2. Experiments show improvements under three and five degradation settings, with significant advantages in tasks such as deraining, dehazing, and deblurring.

Weaknesses:

1. The auxiliary supervision definition of degradation type t and degradation intensity s in the proposed method is not clear enough. The paper does not clearly explain how to consistently define and normalize the degradation intensity in heterogeneous tasks such as denoising, deraining, dehazing, deblurring, and low-light enhancement, nor does it explain how to handle degradation in mixed degradation cases.

2. Under the soft-tagged setting, the actual gain of TTD is quite limited, only 32.31-32.33dB, and the gap between hard and soft tags only slightly decreases from 0.57dB to 0.50dB. This indicates that its benefits for unlabeled inference are relatively limited.

3. The proposed method mainly targets spatial non-uniform degradation, but the crucial ablation experiments were only conducted on the denoising task. It is recommended that the authors provide additional ablation experiments for deraining, defogging, and deblurring components to more robustly validate the effectiveness of the proposed module.

---

> ### Author Rebuttal · Authors · 2026-03-31
>
> We would like to thank the reviewer for the constructive feedback. We are glad that the reviewer finds our paper technically clear with a well-defined design goal addressing the limitations of fixed patch tokenization, and the experiments showing significant improvements across both three and five degradation settings. We address the concerns as follows.
>
> **(1) Clarification of `t` and `s`, and how mixed degradations are handled.**
> In FIT, `t` and `s` are **training-only auxiliary supervision** for DGDE; they are **not required at inference**. At test time, FIT takes only the degraded image as input. `t` is the degradation-type label for the auxiliary classification branch. `s` is a **task-wise normalized scalar severity target** for the auxiliary regression branch. We do **not** impose one absolute severity scale across heterogeneous tasks. Instead, for each degradation family, we normalize its generation parameter into `[0,1]`, so `s` preserves relative severity **within** each task.
>
> |Task|`t`|`s`(normalized to `[0,1]`)|
> |---|---|---|
> |Denoise|noise|noise level `σ/σ_max`|
> |Derain|rain|rain-streak intensity / max training intensity|
> |Dehaze|haze|scattering coefficient `β/β_max`|
> |Deblur|blur|motion trajectory length / max training length|
> |Low-light|low-light|illumination reduction ratio / max training reduction|
>
> For deblurring and low-light, these quantities come directly from the synthetic generation process. For **mixed degradations**, we do **not** define a single cross-task severity target or require a unique mixed label at inference. Instead, mixed cases are handled by the **image-derived degradation representations** `(g,M)` together with robust task conditioning.
>
> In summary, `t/s` are only auxiliary supervision for DGDE during training; mixed cases at test time are handled from the degraded image itself. We will make this protocol explicit in the revision and add the exact normalization formulas.
>
> **(2) Why TTD matters although the standard soft-token gain is modest.**
> We agree that the gain under the standard single-task soft-token setting is modest. The main purpose of TTD is to improve robustness when the task cue is **missing, ambiguous, or incorrect**, rather than to boost the already-easy single-task case. To verify this, we design a wrong-token robustness test: for each test image with ground-truth task label k, we run inference K−1 times, each time substituting a different incorrect task token from the remaining K−1 tasks. The "Avg. wrong token" column reports the PSNR averaged over all images and all mismatched assignments:
>
> |Setting|Correct token|Soft token|Avg. wrong token|
> |---|---:|---:|---:|
> |w/o TTD|32.88|32.31|32.07|
> |TTD(`p_drop`=0.1)|32.86|32.34|32.15|
> |TTD(`p_drop`=0.3)|32.83|32.33|32.25|
>
> Without TTD, the PSNR drop from correct to wrong tokens is **0.81dB**, indicating strong reliance on the oracle task label. With TTD(`p_drop`=0.3), this drop is reduced to **0.58dB**, confirming that TTD makes the model substantially less sensitive to incorrect task labels. Combined with the mixed-degradation improvement already shown in Table 4 (+0.69dB), these results demonstrate TTD's intended role: improving robustness when the task identity is unreliable or mixed. TTD is training-only and introduces **zero inference overhead**.
>
> **(3) Ablation beyond denoising and evidence for spatially non-uniform degradations.**
>
> In the main text, we adopt denoising because it provides the most explicit control to the severity sweep for grid-artifact analysis, but the mechanism is not specific to denoising. We supplement the same cumulative ablation on deraining, dehazing, and deblurring. The baseline (vanilla Transformer without any proposed component) corresponds to Row A in Table 3; the final score matches the full-model result in Table 2:
>
> |Task(Benchmark)|+Deform|+Deg-Cond|+Seam|Full FIT(+TTD)|
> |---|---:|---:|---:|---:|
> |Derain(Rain100L)|37.41|38.02|38.34|38.56|
> |Dehaze(SOTS)|29.98|30.34|30.55|30.81|
> |Deblur(GoPro)|28.87|29.12|29.31|29.52|
>
> The trend is highly consistent across all three tasks. **Deform** gives the largest gain, showing that adapting tokenization/reconstruction boundaries is especially beneficial when fixed patch grids cut across rain streaks, haze-depth transitions, or motion-blurred edges. **Deg-Cond** adds a stable gain, showing that degradation-aware conditioning improves where and how the boundaries should bend. **Seam** and the final **TTD** stage then provide smaller but consistent gains, indicating that smoother reconstruction and more robust task conditioning further strengthen the same mechanism.
>
>
> Thanks for the valuable comments again. We will include all the above discussions and results in the revision. We sincerely hope that our responses alleviate the reviewer's concerns.

---

> > ### Author Rebuttal · Reviewer_isRs · 2026-04-05
> >
> > My problem has been solved. I raised my score to weak accept.

---

> > > ### Author Response · Authors · 2026-04-07
> > >
> > > We sincerely thank you for re-evaluating our work and for the constructive suggestions that helped improve the paper. We will incorporate all discussed revisions in the final version.

---

### Decision · Program_Chairs · 2026-04-30

**Decision:**

Accept (regular)

**Comment:**

Across all reviewers, there is a consistent agreement that the paper is technically sound, clearly motivated, and empirically effective. The proposed method is well formulated, with a coherent pipeline integrating degradation estimation, conditioning, and deformable sampling. Experimental results demonstrate consistent improvements over baselines across multiple restoration tasks, and provide evidence that the approach alleviates grid artifacts.

At the same time, reviewers raised several concerns (I think they are reasonable and important):

1. Limited conceptual novelty: While the integration of degradation-aware conditioning with deformable tokenization is meaningful, the core components are closely related to existing techniques such as deformable convolutions and attention. The contribution is therefore better characterized as a solid architectural refinement rather than a fundamentally new paradigm.

2. Clarity and formulation issues: Some aspects, such as the definition of degradation type/intensity, seam loss formulation, and visualization of artifact reduction, were initially unclear. These issues were largely addressed during rebuttal, leading reviewers to maintain or raise their scores to weak accept.

3. Evaluation limitations: Reviewers noted missing analyses, including efficiency (parameters/FLOPs/latency), broader ablations (beyond denoising), stronger baselines, and experiments on mixed degradations or cross-domain generalization. These would strengthen the empirical validation but are not critical flaws.

Importantly, after the rebuttal phase, all reviewers converged to positive (weak accept) recommendations, indicating that the main concerns were either resolved or considered non-blocking. Overall, this paper presents a well-motivated and carefully executed piece of work that offers a meaningful improvement to unified image restoration frameworks. While the level of novelty is moderate and some empirical analyses could be further strengthened, the technical soundness and consistent performance gains make it a contribution to the field.